



# Ice nucleating particle concentrations of the past: Insights from a 600 year old Greenland ice core

Jann Schrod[1], Dominik Kleinhenz[1], Maria Hörhold[2], Tobias Erhardt[3], Sarah Richter[1],
Frank Wilhelms[2,4], Hubertus Fischer[3], Martin Ebert[5], Birthe Twarloh[2], Damiano Della Lunga[2], Camilla
Marie Jensen[3], Joachim Curtius[1], and Heinz G. Bingemer[1]

[1]Institute for Atmospheric and Environmental Sciences, Goethe University Frankfurt, Frankfurt am Main, Germany
[2]Alfred-Wegener-Institut Helmholtz-Zentrum für Polar-und Meeresforschung, Bremerhaven, Germany
[3]Climate and Environmental Physics, Physics Institute & Oeschger Centre for Climate Change Research, University of Bern,
Bern, Switzerland
[4]GZG, Department of Crystallography, University of Göttingen, Göttingen, Germany
[5]Institute for Applied Geosciences, Technical University of Darmstadt, Darmstadt, Germany

**Correspondence:** Jann Schrod (schrod@iau.uni-frankfurt.de)

**Abstract.**

Ice nucleating particles (INPs) affect the microphysics in cloud and precipitation processes. Hence, they modulate the radiative properties of clouds. However, atmospheric INP concentrations of the past are basically unknown. Here, we present INP measurements from an ice core in Greenland, which dates back to the year 1370. In total 135 samples were analyzed with the

FRIDGE droplet freezing assay in the temperature range from $-14\,°C$ to $-35\,°C$. The sampling frequency was set to 1 in 10 years from 1370 to 1960. From 1960 to 1990 the frequency was increased to 1 sample per year. Additionally, a number of special events were probed, including volcanic episodes. The typical time coverage of a sample was on the order of a few months. Historical atmospheric INP concentrations were estimated with a conversion factor, which depends on the snow accumulation rate of the ice core, particle dry deposition velocity and the wet scavenging ratio. Typical atmospheric INP concentrations were

on the order of $0.1\,L^{-1}$ at $-25\,°C$. The INP variability was found to be about $1-2$ orders of magnitude. Yet, the short-term variability from samples over a seasonal cycle was considerably lower. INP concentrations were significantly correlated to chemical tracers derived from continuous flow analysis (CFA) and ion chromatography (IC) over a broad range of nucleation temperatures. The highest correlation coefficients were found for the particle concentration ($d_p > 1.2\,µm$). The correlation is higher for the seasonal samples, where INP concentrations follow a clear annual pattern, highlighting the importance of the

annual dust input in Greenland from East Asian deserts during spring. Scanning electron microscopy (SEM) of single particles retrieved from selected samples found particles of soil origin to be the dominant fraction, verifying the significance of mineral dust particles as INPs. Overall, the concentrations compare reasonably well to present day INP concentrations, albeit they are on the lower side. However, we found that the INP concentration at medium supercooled temperatures differed before and after 1960. Average INP concentrations at $-23\,°C$, $-24\,°C$, $-25\,°C$, $-26\,°C$ and $-28\,°C$ were significantly higher (and more

variable) in the modern day period, which could indicate a potential anthropogenic impact or some post-coring contamination of the topmost, very porous firn.



# 1 Introduction

Ice cores offer an unparalleled opportunity to study atmospheric conditions of the past. The physico-chemical state of the atmosphere is recorded and preserved for many atmospheric constituents in the form of entrained gas bubbles and aerosol particles. Fittingly, ice core archives have been dubbed *a window into the past* by scientists and media alike. Over the past

decades, many different parameters and proxies have been measured in ice cores using a diverse set of techniques from a variety of scientific fields. Unfortunately, the very property that enables most of atmospheric ice formation in the first place, i.e. heterogeneous ice nucleation, has not received much attention with regards to ice core studies. As of today, there has been only a single study to analyze the concentration of ice nucleating particles (INPs) from ice cores (Hartmann et al., 2019). Yet, this parameter is vitally important from an atmospheric science and climate-modelling perspective, since it strongly

influences cloud formation and modulates precipitation processes. Mülmenstädt et al. (2015) confirmed that the majority of global precipitation is produced in clouds involving the ice phase. Moreover, INPs influence the phase of a cloud and in turn interact with radiation processes (Lohmann, 2015).

There is a number of open questions regarding the nature of INPs: their geographical and vertical distribution, seasonal variation, as well as the types of aerosol particles that contribute to their population even in today's atmosphere. Hence, it

is hardly surprising that very little knowledge exists about the pre-industrial concentrations and sources of INPs. Carslaw et al. (2017) reviewed the state of aerosols in the pre-industrial atmosphere. However, they did not discuss this specific matter in depth, because "our understanding of global ice-nucleating particles in terms of particular aerosol components is only just emerging" (Carslaw et al., 2017). They conclude that a potential change in INP concentrations since the pre-industrial period remains entirely open, as are related impacts on cloud formation, precipitation processes and the radiation budget.

Albeit, knowledge about the pre-industrial baseline of cloud-active aerosols is essential for climate modelers, as – together with the scattering and absorption properties of the aerosol – it defines the baseline upon which the current radiative forcing by anthropogenic aerosols is calculated. In fact, climate models are highly sensitive to the pre-industrial aerosol conditions (Andreae et al., 2005; Carslaw et al., 2013). As a consequence, a lack of knowledge of the pre-industrial aerosol state leads to large uncertainties when radiative forcings are estimated.

Many modeling studies exist, which suggest that general aerosol characteristics have changed significantly since pre-industrial times, such as aerosol composition, number concentration, size distribution and mixing state (e.g. Stier et al., 2006; Tsigaridis et al., 2006; Hamilton et al. , 2014). Although usually not as straightforward, this trend can be seen in ice core observational data for black carbon (McConnell et al., 2007; Kaspari et al., 2011) and various other trace elements and aerosols (Kaspari et al., 2009; Carslaw et al., 2017, and references therein). Yet, the literature consensus indicates that most anthro-

pogenic aerosol particles are typically poor INPs. For example, Chen et al. (2018) found that the heavy air pollution of Beijing did not affect the INP concentration in this urban setting in the investigated temperature range from $-6\,°C$ to $-25\,°C$. However, this does not per se mean that the INP population as a whole has not changed at all over the last centuries. On the contrary, it seems rather likely that certain particles with ice nucleating potential may in fact be more abundant in today's atmosphere. Biomass burning aerosol is probably the most uncertain and least likely contributor here. Both the magnitude and sign of a





potential anthropogenic change in global fire emissions since the pre-industrial period is the subject of active scientific debate (Hamilton et al., 2018, and references therein). Moreover, literature is still split about the ice nucleating capability of aerosol particles from biomass burning (Twohy et al., 2010; McCluskey et al., 2014; Umo et al., 2015; Levin et al., 2016). An-

thropogenic metal enriched particles from industrial processes (e.g. from coal combustion, mining, smelting, etc.) have been consistently found in the Arctic during Arctic Haze events (e.g. Shaw, 1995). Such heavy metals appear regularly as a small fraction of ice residuals in field experiments (Ebert et al. , 2011; Eriksen Hammer et al. , 2018). The most likely candidate for INPs, however, may actually be soil or desert dust from areas that have been subject to land-use change and desertification since the pre-industrial times. The IPCC Special Report Climate Change and Land (IPCC, 2019) estimates that $12-14\%$ of today's global ice-free land surface are croplands. Intensive and extensive pasture land as well as savannahs and shrublands

used for livestock farming make up another $30-47\%$. Due to the expansion of these agricultural areas as well as the practices themselves, the erosion of these agricultural soils is increasing. In fact, it is estimated that soil erosion is currently between $10-100$ times higher (depending on tillage) than the natural soil formation rate (IPCC, 2019). Eroded soil particles may enter the atmosphere and potentially affect micro-physical cloud formation processes. Indeed, (agricultural) soil dust has been proven to be an active INP in many studies (Conen et al., 2011; Tobo et al., 2014; O'Sullivan et al., 2015). Furthermore, the range and

intensity of desertification, which is defined as the land degradation of arid, semi-arid and dry sub-humid areas (i.e. drylands), has increased in the past several decades (Shukla et al., 2019). According to the IPCC Special Report, drylands currently cover about 46% of the global land area, about 9% of which were identified as hotspots of desertification. Especially during droughts such areas are susceptible to higher dust storm activity, which may introduce more mineral dust to the atmosphere. For example, Ganor et al. (2010) found the number of events with transport of African dust over the Eastern Mediterranean to increase

significantly over the years $1958-2006$. Concerning Greenland, isotopic studies showed that the main sources of mineral dust aerosol in both glacial times and during the Holocene are natural Chinese desert areas, and in particular the Taklamakan desert (Svensson et al., 2000; Bory et al., 2003), however a recent anthropogenic increase in mineral dust concentration has not been found.

Hartmann et al. (2019) were the first to estimate the concentration of INPs from two Arctic ice cores from Lomonosovfonna,

Svalbard (78.82° N, 17.43° E) and Summit, Greenland (72.58° N, 37.64° W) using the droplet freezing devices LINA and INDA. They analyzed 69 samples in total (42 from Svalbard and 27 from Greenland). Svalbard samples were subdivided into multi-year samples (6 samples in $2-6$ year resolution) and sub-year samples (36 samples with a resolution of some months), which covered the same time periods as the multi-year samples. The investigated time periods were approximately 1480, 1720, 1780, 1800 and 1950. Greenland samples covered the time from 1735 to 1989 with a resolution of about $2-5$

years per sample. INP analysis is presented mainly for T = $-10\,°C$, $-15\,°C$ and $-20\,°C$. Hartmann et al. (2019) observed no long-term trend in the INP concentration. Furthermore, they found the "short-term" variability of INP concentrations from adjacent sub-year samples to dominate the total variability of the complete data set. In their closing remarks, they recommend future studies to focus on creating a continuous record of ice core INP concentrations for the last few centuries, to include a simultaneous analysis of INP-related chemical and biological substances, and to analyze ice cores from other Arctic locations

to gain knowledge about the spatial distribution of INP concentrations over time.



Here, we present INP data from an ice core from north eastern central Greenland (B17, 72.25° N, 37.62° W) that dates back to about 1370 (Weißbach et al., 2016). The ice core was drilled in the framework of the North Greenland Traverse (NGT, 1993 – 1995) and reaches a depth of about 100 m. In total 135 samples were measured with the FRIDGE instrument (Klein et al., 2010; Schrod et al., 2016) in its droplet freezing mode (Hiranuma et al., 2015; DeMott et al., 2018; Hiranuma et al., 2019).

Samples were selected in regular intervals of 10 years to cover the whole period of the ice core. Most of these discrete samples typically integrate over a time period of half a year. Furthermore, sampling frequency was increased to 1 sample per year between 1960 – 1990 to establish a statistically sound "modern day" reference period. Moreover, certain samples were selected based on extraordinarily high concentrations in their "dust" concentration and/or peaks in the signal of conductivity, which is a measure of the acidity of the atmosphere (i.e. from volcanic eruptions). Prior to our ice nucleation measurements, the ice core

was analyzed for dust, conductivity, and soluble particle concentrations of $Ca^{2+}$, $Na^+$, $NH_4^+$ and $NO_3^-$ using continuous flow analysis (CFA) (Kaufmann et al., 2008) at the University of Bern and for $Ca^{2+}$, $Na^+$, $NH_4^+$, $NO_3^-$, $K^+$, $Mg^{2+}$, $F^-$, $MSA^-$, $Cl^-$, $Br^-$ and $SO_4^{2-}$ using ion chromatography (IC) at the Alfred-Wegener Institute for Polar and Marine Research, Bremerhaven. Finally, we analyzed a high resolution period with almost monthly resolution (1463 – 1464, N = 12) to investigate a potential seasonal variation in the INP concentrations.

## 2 Methods

### 2.1 Ice core NGT B17

The ice core B17 was drilled during the NGT in 1993 – 1995 by the Alfred-Wegener-Institute as one of 13 ice cores along the traverse. The drill site is located east of the main ice divide in Central Greenland (72.25° N, 37.62° W, 2820 m asl). The ice core has a total depth of 100.8 m, a depth which corresponds to the year 1363 CE. More information about the characteristics

of B17 and the other NGT ice cores can be found in Weißbach et al. (2016). Weißbach et al. (2016) presents and discusses the density profile, the water accumulation rate, and the ratio of stable water isotopes ($\delta^{18}O$) of the ice cores B16 – B23 and B26 – B30, while chemical profiles of selected NGT ice cores are presented in Fischer et al. (1998) and Bigler et al. (2002). Recently, Burgay et al. (2019) introduced $Fe^{2+}$ as a potential new proxy to identify volcanic events by presenting B17 measurements of a chemiluminescence method.

The ice core was dated on the basis of identified volcanic layers (Weißbach et al., 2016), interpolating between these tie points, supported by measurements of stable water isotopes.

### 2.2 Sample preparations and overview of measurements

After the B17 core was drilled in the early 1990s, it was cut into pieces of 1 m each and stored at −25 °C in a cold storage at the Alfred Wegener Institute for Polar and Marine Research (AWI) in Bremerhaven. In 2018, a longitudinal subsection

of the ice core with a cross section of 35 mm x 35 mm was cut from the whole length of the core. The pieces of the ice core were transported in frozen state to the University of Bern, where they were continuously melted using a well-established





decontamination technique, which is the first step in the continuous flow analysis (CFA) (Kaufmann et al., 2008) (section 2.3). The decontaminated meltwater flow was then split between the online chemical analysis, and a fraction, which was sampled directly into numbered clean vials of discrete aliquots for offline ion chromatography (IC) measurements (section 2.4). Each vial was filled with approximately $1.5 - 8\,\mathrm{mL}$ of sample water, thus covering $4 - 20\,\mathrm{cm}$ of core depth. Depending on the exact

sampling resolution this corresponds to a time resolution of about $1 - 10$ data points per year. The vials were refrozen, shipped to AWI and subsequently measured for the concentration of a large number of major ions, expanding on the CFA measurements. As the IC analysis required a few µL only, the vials still contained most of the sample water after the measurements. Some of these samples were later selected for the ice nucleation analysis in this study (section 2.5). These samples were transported from Bremerhaven to Frankfurt in a small insulated PP-foam cooling box with additional cold packs to guarantee that the

temperature of the samples was always well below $0\,^{\circ}\mathrm{C}$. Then, samples were stored inside a freezer (WAECO Coolmatic CF-40) at about $-17\,^{\circ}\mathrm{C}$. Temperature variability ranged from $-16.1\,^{\circ}\mathrm{C}$ to $-17.9\,^{\circ}\mathrm{C}$ over a period of 15 hours. Hence, it was ensured that samples remained frozen at all times. Prior to the ice nucleation measurement (section 2.6), the respective sample was slowly melted over night in a refrigerator at about $6\,^{\circ}\mathrm{C}$. After the ice nucleation analysis the samples were refrozen once again. Selected samples that were previously analyzed for their INP activity, were transported to TU Darmstadt for chemical

and morphological single particle analysis using a scanning electron microscope (SEM) (section 2.7).

## 2.3    Online chemical analysis

The longitudinal subsections of the ice core were melted continuously in a way that separates meltwater from the potentially contaminated outside of the ice from the clean meltwater from the inside during the CFA analysis. This decontamination technique is absolutely effective and even gastight for solid ice (density $> 0.82\,\mathrm{kgL^{-1}}$). However, we cannot completely exclude

some minor contamination for the relatively porous firn, which is found at the top of the ice core, as it is possible that melt water is drawn upwards by capillary forces and that surface contaminants may be mixed in here.

     The fraction of the clean meltwater stream is used to feed a range of detectors to determine the concentrations of major ions in the water using purpose-build spectrophotometric methods ($Ca^{2+}$, $Na^{+}$, $NH_4^{+}$ and $NO_3^{-}$), electrolytic conductivity and the concentration of insoluble particles ($> 1.2\,\mu\mathrm{m}$) as detailed in Kaufmann et al. (2008) and references therein. Additionally,

trace-elemental concentrations in the meltwater where determined using online ICP-TOF-MS (Erhardt et al. , 2019), which were, however, not used for this manuscript. All of these measurements are performed continuously during the melting of the ice-core section to produce continuous high-resolution record. Typical analytically limited resolutions are in the range of $0.5 - 1.0\,\mathrm{cm}$ governed by the smoothing of the individual detection methods.

## 2.4    Ion chromatography

The vials containing the discrete decontaminated aliquots of the ice core were melted at room temperature prior to the IC analysis. The vials are then placed into an autosampler to be measured. Simultaneous analysis of anions and cations was performed using a 2 channel Dionex ICS 5000+ Reagent-free HPIC system (Thermo Fischer). For the anion (cation) determination the Dionex IonPac AG18-Fast-4 µm guard column (IonPac CG12A-5 µm column) and the Dionex IonPac As18-Fast-4





μm analytical column (IonPac CS12A-5 μm column) were used. The measurements were calibrated with 7 internal standards, prepared from available anion and cation standards. Pre-mixed external standards are used for quality control. The IC provides a quantitative analysis of the concentration of $Ca^{2+}$, $Na^+$, $NH_4^+$, $NO_3^-$, $K^+$, $Mg^{2+}$, $F^-$, $MSA^-$, $Cl^-$, $Br^-$ and $SO_4^{2-}$.

## 2.5 Sample selection

Figure 1a shows the temporal distribution of the selected samples for the ice nucleation experiment (total N = 135, 7% of all samples) throughout the ice core. We placed a strong emphasis on having a consistent data set with samples in regular time intervals. Time steps of 10 years were decided to be both meaningful and feasible. Furthermore, we were interested in investigating the question whether the pre-industrial INP concentration is different from the INP concentration of the recent past (1960 – 1990). Therefore, the sample frequency was increased to about 1 sample per year in the latter time period.

Moreover, we selected some samples according to peak values in the high resolution CFA measurements. Samples labeled "dust event" include a respective sample that featured a peak in the high resolution signal of particles with spherical diameters larger than 1.2 μm, as well as a couple samples before and after it. Due to the episodic nature of such an event and the automatic collection of sample water into the vials, which averages over peak-independent periods of time, the sample containing the high resolution peak signal must not necessarily have an extraordinarily high average value itself. Similarly, peak samples in the

high resolution signal of conductivity were selected that are typically derived from high sulfuric acid deposition in the ice after volcanic eruptions. One group of these "volcanic event" samples can be unequivocally ascribed to the Laki eruption in 1783/84. The eruption of Grímsvötn (Laki) is the best-characterized historical large volume basaltic fissure eruption in Iceland. The 8-month lasting Laki eruption occurred from a 27 km long volcanic fissure in the Grímsvötn volcanic system. It emitted ≈ 15 km³ of lava, 0.4 km³ of tephra, an estimated 122 Tg of $SO_2$, and other gases and trace metals (Thordarson and Self, 1993,

2003; Sigl et al., 2018). Lastly, 12 adjacent samples with a near monthly resolution were measured in order to estimate the "short-term" variability of the data set and to check for a possible seasonal variation of the INPs.

  Figure 1b gives the temporal coverage of each individual analyzed sample. The representative time of a certain sample is given by its lengths, which is estimated by calculating the average time difference between the sample before and after it. 63% of the analyzed samples average over a time period of 6 ± 2 months. 26% of the samples average over a shorter and 11% over

a longer time period.

## 2.6 Ice nucleation analysis

The ice nucleation measurements were performed with the FRIDGE instrument (Klein et al., 2010; Schrod et al., 2016). FRIDGE hast two operational modes. In its "standard mode" aerosol particles activate ice crystal growth by diffusion of water vapor at supercooled temperatures and near vacuum conditions (Schrod et al., 2017; DeMott et al., 2018; Thomson et al., 2018;

Marinou et al., 2019). In this manuscript however, we solely used FRIDGE in the droplet freezing mode (Hiranuma et al., 2015; DeMott et al., 2018; Hiranuma et al., 2019). The performance of the FRIDGE instrument was tested during the Fifth International Workshop on Ice Nucleation – part 2 (FIN-02, DeMott et al., 2018). In this large-scale laboratory campaign 21 different INP counters were intercompared at the Aerosol Interaction and Dynamics in the Atmosphere (AIDA) facility of the





Karlsruhe Institute for Technology (KIT). FRIDGE agreed generally very well with the other instruments (especially for the immersion freezing method) for the various investigated aerosol types (including natural mineral dusts, dust components and a biological material).

During a measurement of an ice core sample 65 droplets of 2.5 µL ± 5% each are placed homogeneously at random onto a clean silanized Si-substrate of 47 mm diameter on the cold table in the FRIDGE chamber. The droplets are picked up directly from the sample vial and are semi-automatically dispensed using an Eppendorf Multipipette E3 with fresh Eppendorf tips of the highest level of purity (Combotips advanced, biopur grade, 0.1 mL). Temperature is decreased quickly at first in the range from 14 °C to 0 °C and then slowly at a constant cooling rate of 1 °C per minute until every droplet is frozen. The temperature ramp is implemented by a PID-controlled Peltier element that is supported by a cryostat (Lauda, Ecoline Staredition RE110;

ethanol coolant) dissipating the heat. Temperature is controlled by a PT-100 sensor (precision ± 0.2 °C), which is attached to the surface of the wafer. The measurement cell is continuously flushed with synthetic air at a flow of 1 L min$^{-1}$ to prevent condensation and riming and to limit contamination from the laboratory environment during the measurement. The freezing of droplets is observed by a CCD-camera (AVT Oscar F-510C 2/3″). A droplet changes its brightness significantly during the freezing event: droplets are nearly translucent when liquid but are opaque when frozen. LabView software automatically

detects the moment of freezing, records the corresponding temperature and saves the images every ten seconds. The final results are always double-checked by the operator. After all droplets are frozen, the substrate is heated up, cleaned and the process is repeated twice with new droplets. In total, the freezing temperatures of 195 individual droplets are determined for each ice core sample (3 runs with 65 droplets). The cumulative INP concentration per mL of meltwater as a function of temperature ($N_{\mathrm{INPice}}(T)$) is calculated by the well-established Vali (1971) equation:

$$N_{\mathrm{INPice}}(T) = \frac{1}{V_{\mathrm{d}}} \cdot \left[ \ln\left(n_{\mathrm{total}}\right) - \ln\left(n_{\mathrm{total}} - n_{\mathrm{frozen}}\right) \right].$$    (1)

Here, $V_{\mathrm{d}}$ is the volume of each droplet, $n_{\mathrm{frozen}}$ is the number of droplets that are frozen at temperature $T$ and $n_{\mathrm{total}}$ is the total number of droplets of each freezing experiment. With the aforementioned experimental values for these variables we are typically able to resolve INP concentrations between 2 and 2000 mL$_{\mathrm{ice}}$$^{-1}$.

During the nature of the experiment, droplets will freeze in a different temperature range, leading to a slightly inconsistent

data set. This means that while some samples will have measurement data at a warm temperature, others will not. In terms of calculating average concentrations etc. we set the concentration of those non-active samples to zero, knowing that the actual concentrations is somewhere between zero and the lowest resolvable concentration (2 per mL$_{\mathrm{ice}}$). The same issue arises at lower temperatures. For some samples all droplets will be frozen before reaching a certain temperature. In those cases the concentration was set to the highest resolvable concentration (2000 per mL$_{\mathrm{ice}}$). It should be noted that for some cases the

real INP concentration might actually be substantially higher due to the exponential behavior in ice nucleation. The presented average concentrations (section 3) should therefore be considered as a lower estimate.





## 2.7 Scanning Electron Microscopy

We selected three individual samples (years: 1977, 1680 and 1630) to be analyzed with scanning electron microscopy (SEM) to gain information about the chemical signature of deposited aerosol particles. By use of a Quanta 200 FEG Environmental scanning electron microscope equipped with an EDAX Genesis energy dispersive X-ray microanalysis system (EDX) the

elemental composition of individual aerosol particles as well as a rough size-distribution can be determined.

For this purpose, the samples were melted and subsequently filtered using a $25\,\mathrm{mm}$ Nucleopore membrane filter with $0.4\,\mathrm{\mu m}$ pore size and a filter flask at vacuum provided by a water jet. The area of filtration was about $5\,\mathrm{cm^2}$, ensuring that particles were concentrated on a relatively small area, which is advantageous to the SEM analysis.

For SEM-EDX analysis on each filter some 100 rectangular fields of about $100\,\mathrm{\mu m}$ x $100\,\mathrm{\mu m}$ were scanned and for all de-

tected particles the size was determined and an EDX analysis was performed. Using this procedure particles down to approximately $250\,\mathrm{nm}$ will be detected. Smaller particles will often be overlooked, what will be also true for even larger carbonaceous particles, which have a bad contrast on the polycarbonate filter.

## 2.8 Background freezing and uncertainties related to the INP measurements

There are numerous possible sources of contamination in a typical droplet freezing assay, which may cause a droplet to freeze

before the homogenous limit. The potential contamination due to the analytical procedure may be evaluated by establishing a "background" from the freezing of "pure" water droplets. Polen et al. (2018) review the state of the art of droplet freezing techniques, summarize potential contamination sources, and advise on how to report background freezing. In general, contamination may arise from particles in the "pure" water itself, substrate interferences, or the environment that is in contact with the droplets. However, it is rather difficult to assess how much each of these categories contributes to the freezing spectrum of a

specific background sample (Polen et al., 2018). We will now look at each of these factors and describe how exactly they relate to the actual ice nucleation experiment of this manuscript.

First, the surface of the substrate onto which the droplets are placed can induce freezing. It is well known that the contact angle of the droplets influences the freezing process. Furthermore, microscopical cavities, scratches, cracks or other surface defects, as well as actual particles on the surface of the substrate may affect the freezing temperature of the droplets. In

our setup, we use custom-cut silicon wafer as substrates. The wafers were regularly coated with dichlorodimethylsilane in a vacuum desiccator to create a thin hydrophobic layer on the surface of the substrate. The silanization has several positive effects. First, it "seals off" microscopical surface defects on the wafer. Second, the hydrophobic layer prevents condensation and rime formation and thereby limits the effects of a possible Wegener-Bergeron-Findeisen process. The Si-substrates are stored in PetriSlide containers before use. Finally, the surface of a wafer is cleaned thoroughly by hand with pure non-denatured ethanol

(Rotipuran, >99.8%, Carl Roth) immediately before and after each measurement run.

Obviously, the environment surrounding the sample may affect its freezing temperature. Specifically, the vials in which the water is stored and the tips of the pipette may introduce contamination. Furthermore, the droplets can collect particles from laboratory air during a measurement, which may nucleate ice artificially. Our measurement cell is continuously flushed with



particle-free, dehumidified synthetic air at a constant flow rate of 1 L min$^{-1}$. This largely prevents the droplets to come in contact with aerosols from the laboratory and inhibits water condensation and growth of frost.

Lastly, we consider the pure water itself, which is used as a medium to establish the background signal, as a source for contamination. No matter how "pure" a manufacturer certifies its water to be, at very cold temperatures even a single con-
taminant particle inside a droplet might initiate the ice nucleation process and causes the droplet to freeze. For purposes of background measurements we used the "pure" water Rotipuran ultra (Carl Roth). However, in the analysis of droplets from the ice core the samples are not in contact with the "pure" water. Unfortunately, this results in an intrinsic problem of the background evaluation of our ice core measurements. Depending on the relative importance of the role of the reference water in the background measurement, our background freezing spectrum is more or less representative for the ice core freezing spectra. In
other words, if non-water contamination effects dominate the freezing in the background measurement, the respective freezing spectrum could be adapted for the ice core measurements. If the majority of droplets in a background measurement freeze, however, due to contaminants in the reference water, the background freezing curve only serves as an upper limit, meaning that the ice core measurements have an unspecified better background than what the background freezing spectrum would suggest. In fact, we observed some samples from the ice core to freeze even later than our typical background measurements, which
suggests that at least some contamination is introduced by the ultrapure water itself. Therefore, we chose not to subtract the background freezing spectrum from our measurements as is common practice, but give a range of temperatures, where no or only little interference is expected due to background contamination. Figure 2 shows a typical background freezing spectrum compared to the average freezing spectrum of the ice core measurements. Accordingly, results of freezing temperatures colder than $-31\,°C$ are likely to be influenced by a reasonable amount of background freezing (frozen fraction $\geq 25\%$) and should
be interpreted with care. Only little background influence is expected, however, for temperatures warmer than $-28\,°C$ (frozen fraction $\leq 10\%$).

Moreover, we tested whether the freezing temperature of an individual droplet can be reproduced. For 120 (4 x 30) droplets the freezing temperatures $T_1$ and $T_2$ were individually measured during two subsequent freezing cycles (Fig. 3). Overall, we found that over a wide range of temperatures the freezing temperature is a property of each individual droplet and can be
reproduced fairly accurately. The temperature difference between two subsequent freezing cycles of the same droplet was below $\pm 0.5\,°C$ for 79% of the cases. Half of the droplets showed a temperature difference of $\leq 0.23\,°C$. Only 5 of 120 droplets differed by more than $\pm 1\,°C$ in their nucleation temperature. Furthermore, the resulting slope of the data was close to unity (1.016) with a strong linear correlation ($R^2 = 0.96$). These results suggest that a) temperature uncertainty in FRIDGE is relatively low, b) repeated cycles of freezing and defreezing of an ice core sample do not affect its ice nucleation properties
and c) this test sample showed a mostly deterministic ice nucleation behavior.

In this manuscript, we specify the uncertainty of the INP concentration as the 95% confidence interval, which is derived from the freezing statistics alone (i.e. number of frozen droplets at a certain temperature). The uncertainty of the INP concentration is high for the very first few drops (i.e. often as high or higher than $\pm 100\%$) and levels out usually below $\pm 20\%$ at lower temperatures, depending on the specific freezing spectrum.





## 2.9 Other uncertainties

There are undoubtedly many difficulties and uncertainties associated with estimating (atmospheric) INP concentrations from an ice core. First of all, high standards of precaution need to be met in order to prevent contamination effects (e.g. when cutting the ice, when handling and storing the samples, or during the INP analysis itself). Second, the overarching question is whether

general source conditions, transport patterns and dry and wet deposition efficiencies, which directly influence the number of particles inside the ice as well as their possible source attribution, can be assumed not to have changed substantially over the time scale covered by the ice core. It is likely that for the time scale of a few centuries, when climate conditions were similar to today, this assumption holds up (Wolff, 1996), although there is obviously some uncertainty and unknown variability.

Moreover, it is often implicitly assumed that aerosol particles in ice core archives perfectly resemble the actual atmospheric

situation at the time the particles entered the ice sheet. Yet, there is a number of possible biases to consider for the case of INPs. There are several routes an aerosol particle could have taken to end up in an Arctic ice core: a) The aerosol particle may have simply been transported to the ground by dry deposition, b) it was activated as a cloud nucleus (either INP or CCN) and was subsequently removed from the atmosphere via precipitation or c) it was removed either by in cloud or below cloud scavenging. This means that aerosol particles with good ice nucleation activity may actually be preferably deposited in an ice

core as compared to particles that are not as ice active (Dibb, 1996). The relative importance of riming processes determines how significant this potential bias is (i.e. the bias is low if most of the particles are transferred to the surface of the ice sheet by riming or dry deposition). Furthermore, INPs are typically large in diameter ($> 0.5\,\mu m$, DeMott et al. (2010)). Dry deposition is more efficient for larger particles (gravitational settling) and for very small particles (Brown diffusion) than for $\sim 100\,nm$ particles, which represent the largest fraction the aerosol population. This means as well that INPs may be overrepresented in

an ice core as compared to non-ice-nucleating-particles or to the surrounding atmosphere.

The next question is whether aerosol particles irreversibly remain inside of an ice sheet and if so, whether these particles stay physically and chemically inert while being preserved in the ice or if they experience modifications. Indeed, non-volatile atmospheric particles are considered to be essentially chemically inert and physically immobile once they are transferred to the ice. Yet, it is basically unknown if or how surface properties (e.g. active sites) of a particle are modified in the ice. Aerosol

particles seem to remain at a given layer throughout the firnification (Bales and Choi, 1996). Aggregation of dust particles has been only observed close to the bottom of the Antarctic ice sheet (Tison et al., 2015). Hartmann et al. (2019) come to the same conclusion that INPs are well preserved in an ice core and a reconstruction of their concentration for past climates is possible.

However, it is likely that aerosol particles will alter physically and chemically during the atmospheric transport from their source region to the Arctic. This includes, but is not limited to, changes in size distribution, mixing state or coatings. We will

therefore not try to speculate about potential atmospheric INP concentrations at a possible source location in the past.

In summary, one should be wary when interpreting results based on aerosol data from an ice core. Statements based on these findings should be assessed carefully. Or in the words of Albrecht Neftel: "The reconstruction of an atmospheric record from the concentration versus depth profile gained from ice cores is similar to an odyssey through a labyrinth with many pitfalls ready to slur over enthusiastic students and researchers" (Neftel, 1996).





In the light of these uncertainties associated with the transfer processes of INPs between the atmosphere and the ice sheet, the conversion factor from in-ice-concentrations to (Arctic) atmospheric concentrations, which will be introduced in the following chapter, should therefore be interpreted only as an order-of-magnitude estimation.

## 2.10 Conversion to atmospheric concentrations

To convert the cumulative INP concentration per volume of meltwater to an atmospheric concentration, we follow the theoretical considerations presented in Fischer et al. (2007). As for any aerosol particle, an INP can be transferred from the air to the surface of the ice sheet either by dry deposition (predominantly gravitational settling and turbulent transport) and wet deposition (cloud particle activation or riming and subsequent removal by precipitation, or below-cloud scavenging). Fischer et al. (2007) states that in a simplified model the total deposition flux $J_{\mathrm{ice}}$ (i.e. the sum of the flux of dry and wet deposition, $J_{\mathrm{dry}}$ and

$J_{\mathrm{wet}}$, respectively) to the ice surface is defined by the product of the snow accumulation rate $A$ and the average concentration of the investigated species (i.e. here for INPs: $N_{\mathrm{INPice}}$) in the ice core sample. Over long periods of time the deposition flux can be written as:

$$J_{\mathrm{ice}} = A \cdot N_{\mathrm{INPice}} = J_{\mathrm{dry}} + J_{\mathrm{wet}} = v_{\mathrm{dry}} \cdot N_{\mathrm{INPatm}} + A \cdot \varepsilon \cdot N_{\mathrm{INPatm}}, \qquad (2)$$

where $N_{\mathrm{INPatm}}$ is the atmospheric INP concentration (or any other investigated species of interest), $v_{\mathrm{dry}}$ is the dry deposi-

tion velocity and $\varepsilon$ is the effective scavenging efficiency including in-cloud and below-cloud scavenging. Experimentally, $\varepsilon$ is often defined as particle concentration in cloud water or in precipitation (snow/ice/rain) divided by the airborne particle concentration. Rearranging Eq. 2 leads to:

$$N_{\mathrm{INPatm}} = \frac{N_{\mathrm{INPice}}}{\frac{v_{\mathrm{dry}}}{A} + \varepsilon}. \qquad (3)$$

Thus, it is possible to calculate the (Arctic) atmospheric INP concentration, when realistic values for the variables $A$, $v_{\mathrm{dry}}$ and

$\varepsilon$ are estimated. However, Eq. 2 implies that if deposition fluxes change over the time span of the ice core (in particular the wet deposition, which is directly related to changes in the precipitation rate), the concentration of the investigated species in the ice will change as well. This means that not all potential changes seen in the ice core INP concentration, are necessarily caused by actual changes in the atmospheric concentration. Henceforth, we will, however, treat these variables as constants due to the lack of a better knowledge and because climate conditions changed only little over the last centuries. In particular, the average

snow accumulation of the B17 ice core has been determined by Weißbach et al. (2016) and shows little variation over time ($A = 11.4 \pm 0.1\,\mathrm{cm\,water\,equivalent\,a^{-1}}$, N = 630). Unfortunately, the other deposition parameters are not as well-known.

In general, $v_{\mathrm{dry}}$ heavily depends on the particle diameter, shape, density and physical properties of the particle. The typical range is between $10^{-2}\,\mathrm{cm\,s^{-1}}$ and $10\,\mathrm{cm\,s^{-1}}$ (Seinfeld and Pandis, 2006). Smaller particles ($d_{\mathrm{p}} < 0.1\,\mathrm{\mu m}$) and larger particles ($d_{\mathrm{p}} > 1\,\mathrm{\mu m}$) usually have higher dry deposition velocities than medium sized particles, where Brownian diffusion and gravi-

tational settling are low (Davidson et al., 1996). Moreover, the nature of the surface itself (e.g. surface type and smoothness) and the level of atmospheric turbulence at the nearest layer to the ground have a major influence on $v_{\mathrm{dry}}$ (Seinfeld and Pandis, 2006). Moreover, over the ice sheet, the dry deposition is strongly influenced by snow ventilation effects induced by





surface roughness (Cunningham and Waddington, 1993). Khan and Perlinger (2017) evaluated five different dry deposition parametrizations with respect to their ability to accurately explain field observations from five land use categories (snow/ice: 8 studies). The parametrization by Zhang and He (2014) performed best overall, and best for snow/ice covered surfaces in particular. Therefore, we used the parametrization by Zhang and He (2014)(Eq. 4) to estimate the dry deposition velocity for

PM$_{2.5}$ aerosol particles. The parametrization is predominantly dependent on the so-called friction velocity $u_*$ and the particle diameter $d_{\mathrm{p}}$. Khan and Perlinger (2017) use a value $u_* = 0.12\,\mathrm{m\,s^{-1}}$ for snow/ice surfaces in their observation based accuracy test evaluation. We decided to set $d_{\mathrm{p}}$ in the parametrization to 0.5 μm, since particles of this size and larger are typically considered to be "good" INPs (DeMott et al., 2010, 2015). Osman et al. (2017) analyzed *modern day* samples from two ice cores from west-central Greenland with a time-of-flight single-particle mass spectrometer to determine the size and composition of

insoluble particles. The median particle diameter of insoluble particles within the detectable aerodynamic size range of $0.2 -$ 3 μm was about 520 nm (mean 595 nm $\pm$ 360 nm, N = 8021), which agrees well with our assumption. Filling in the other variables given in Zhang and He (2014) and Khan and Perlinger (2017), we find a dry deposition velocity of $v_{\mathrm{dry}} = 0.05\,\mathrm{cm\,s^{-1}}$. This value agrees well with the dry deposition velocity of $0.03\,\mathrm{cm\,s^{-1}}$, which is used for all aerosols over snow and ice surfaces in the GEOS-chem model (GEOS-Chem, 2011).

The scavenging efficiency $\varepsilon$ (also known as scavenging ratio or washout ratio) is even less well known than the dry deposition velocity. The scavenging ratio is a very complex parameter that is controlled by the particle's size, its physical shape and chemical composition, as well as by cloud properties such as droplet size, cloud temperature and cloud type, and by the vertical extent of rain and cloud (Duce et al., 1991; Shao, 2008). Hence, accurate predictions of $\varepsilon$ are very difficult (Shao, 2008). Duce et al. (1991) warns that experimentally determined concentrations at the ground do not necessarily have to reflect the condi-

tions near the cloud, where the particles are mainly scavenged. Furthermore, $\varepsilon$ can vary greatly for different particle species and should therefore be assessed carefully (Duce et al., 1991). Attention should also be paid to the fact that $\varepsilon$ is reported in the literature either in a mass- or volume-based dimension (($g_{\mathrm{species}}/g_{\mathrm{precip}}$/)/($g_{\mathrm{species}}/g_{\mathrm{air}}$) vs. (($g_{\mathrm{species}}/\mathrm{cm^{-3}}_{\mathrm{precip}}$)/($g_{\mathrm{species}}/\mathrm{cm^{-3}}_{\mathrm{air}}$)). These two definitions differ by the factor $\rho_{\mathrm{precip}}/\rho_{\mathrm{air}}$ ($\varepsilon_{\mathrm{vol}}$ is about 1000 times higher than $\varepsilon_{\mathrm{mass}}$). Usually, $\varepsilon$ is calculated by measuring the airborne concentration of a species and its concentration in a precipitation sample simultaneously at the ground.

The volume-based scavenging ratio is typically in the range of $10^5$ to $10^6$ (Slinn et al., 1978). Mass-based scavenging ratios for mineral aerosols are typically somewhere between 100 and 2000 (Duce et al., 1991). Davidson et al. (1996) reported Arctic $\varepsilon_{\mathrm{mass}}$ for Ca to be 840 for Summit, Greenland. For the purpose of this manuscript we use a value for $\varepsilon$ that is derived from long time observations by Cheng and Zhang (2017). They measured the scavenging ratio for various species at 13 Canadian stations for several years. They give a long-time average value for several species, each composed of individual means from

months that experienced at least 15 days with more than $0.2\,\mathrm{mm}$ of precipitation. The combined measured concentrations of Ca$^{2+}$, Mg$^{2+}$, and Na$^+$ can be taken as a proxy for coarse particulate aerosols (e.g. mineral dust). The long-time average of all 13 stations of these three species yields $\varepsilon_{\mathrm{mass}} \approx 1.12 \cdot 10^3$, which we will use for the scavenging ratio in this manuscript. Note, that the densities of air and water, which are part of the definition of $\varepsilon$, depend on temperature and altitude. Here, we assumed the densities of air $\rho_{\mathrm{air}}$ and water $\rho_{\mathrm{water}}$ to be $1.01\,\mathrm{kg\,m^{-3}}$ ($-25\,°\mathrm{C}$, $2820\,\mathrm{m}$) and $1000\,\mathrm{kg\,m^{-3}}$, respectively. This yields a $\varepsilon_{\mathrm{vol}}$

of $1.11 \cdot 10^6$. Moreover, technically $\varepsilon_{\mathrm{vol}}$ compares the mass and not the number of a certain species within a volume of water





and air. INP concentrations, however, give the number of ice-active particles per volume. Considering the large uncertainties accompanied with the scavenging ratio, we disregard this inconsistency.

Following these assumptions, we obtain a factor of about $8 \cdot 10^{-7}$ for converting from $N_{\mathrm{INPice}}$ to $N_{\mathrm{INPatm}}$ (Fig. 4, blue cross). Figure 4a displays the range of the possible conversion factors as a result of other combinations of $v_{\mathrm{dry}}$ and $\varepsilon$. Figure 4b

shows the sensitivity of the chosen conversion factor associated with the uncertainties in dry and wet deposition efficiencies. Judging from the typical range of literature values of $v_{\mathrm{dry}}$ and $\varepsilon$, the uncertainty of the conversion factor is likely within $\pm 50\%$ of our best estimate. Likewise, our conversion factor is only about twice as high as the conversion factor proposed in Petters and Wright (2015), who compiled INP data from precipitation measurements and translated these to atmospheric INP concentrations at cloud level. Petters and Wright (2015) based their estimation on the assumption that cloud droplets of

typically $1\,\mathrm{pL}$ (each containing no more than one INP) dispersed in $1\,\mathrm{m}^3$ of air weigh about $0.4\,\mathrm{g}$ (cloud water content (CWC) ranges between 0.2 and $0.8\,\mathrm{g\,m^{-3}}$). Depending on the exact CWC, the uncertainty of the Petters and Wright (2015) estimation is also a factor of 2.

Similar to what we propose here, Schüpbach et al. (2018) successfully implemented the assumptions described above into a trajectory based source apportionment study to translate ice core concentrations of $Na^+$, $Ca^{2+}$, $NH_4^+$, $NO_3^-$ and $SO_4^{2-}$ to

atmospheric source concentrations for a 130k year record of Greenland ice core aerosol data.

## 3 Results and Discussion

Overall, the ice core samples show relatively low ice nucleation activity. Droplets freeze in the range of $-14\,^{\circ}\mathrm{C}$ to $-35\,^{\circ}\mathrm{C}$ (Fig. 2). On average, 1% of the droplets is frozen at $-21.27 \pm 1.43\,^{\circ}\mathrm{C}$, 10% at $-24.93 \pm 1.53\,^{\circ}\mathrm{C}$, 25% at $-26.62 \pm 1.87\,^{\circ}\mathrm{C}$, 50% at $-28.33 \pm 2.09\,^{\circ}\mathrm{C}$ and 90% at $-31.03 \pm 2.02\,^{\circ}\mathrm{C}$, respectively. Reiterating our statements from section 2.8, we find

a reasonable amount of droplets freezing only at temperatures, where some influence from background freezing is expected ($T < -28\,^{\circ}\mathrm{C}$). Accordingly, the interpretation of the lower end of the data is difficult. Figure 5 shows a time series of the fraction of frozen droplets in samples with a regular time interval of 10 years between each sample. In general, this subset of the data (about half of all samples) reflects the basic characteristics of the ice core samples well (see also Fig. S1 in the Supplement), with a few distinct exceptions, which we we will discuss later. The freezing spectra show relatively little variation

between individual samples overall, and from sample to sample. Some notable exceptions are samples that feature an early ice nucleation onset (e.g. 1400, 1430, 1630, 1740, 1950) or a freezing spectrum that is completely shifted to warmer freezing temperatures (e.g. 1450, 1550, 1620, 1630, 1930, 1960, 1990). Of these samples, the sample of 1630 stands out the most. Here, a very steep freezing spectrum was observed with a freezing range of only about $3.5\,^{\circ}\mathrm{C}$ ($-18.27\,^{\circ}\mathrm{C}$ to $-21.74\,^{\circ}\mathrm{C}$). We analyzed this sample a second time to test if our first measurement was caused by some sort of contamination, but the

corresponding freezing spectrum was very similar. This may point to a real atmospheric event. However, CFA and IC data of that sample do not indicate the presence of volcanic particles or exceptional concentrations in any other trace element. Unfortunately, the single particle analysis by SEM did not provide any additional explanation for the remarkable ice activity of this sample (see section 3.2). A contamination during the sampling generation step cannot be excluded altogether.





We turn the discussion now to $1\,^\circ$C-binned average INP concentrations ($\pm$ standard deviation) of all ice core samples. At $-15\,^\circ$C only 3% of the samples (4) showed ice nucleation with an average frozen fraction of only 0.02%. This translates to $N_{\mathrm{INPice}}$ of $0.06 \pm 0.35\,\mathrm{mL^{-1}}$ or $N_{\mathrm{INPatm}}$ of about $5 \cdot 10^{-5}\,\mathrm{L^{-1}}$, respectively. At $-20\,^\circ$C 61 samples (45%) were ice-active, yet still only averaging a frozen fraction of 0.7%. An average in ice INP concentration of $3.06 \pm 18.76\,\mathrm{mL^{-1}}$ was observed, which

corresponds to atmospheric concentrations of about $2 \cdot 10^{-3}\,\mathrm{L^{-1}}$. All samples displayed some freezing activity at $-25\,^\circ$C. The previously mentioned sample from 1630 was the only sample that was completely frozen at this temperature. We found an average in ice concentration of $113.92 \pm 272.01\,\mathrm{mL^{-1}}$, which corresponds to an atmospheric concentration of about $0.09\,\mathrm{L^{-1}}$. Only about 4% of all samples (6) were completely frozen before reaching $-30\,^\circ$C. On average, 69.3% of the droplets were frozen at $-30\,^\circ$C. Here, $N_{\mathrm{INPice}}$ was found to be $668.09 \pm 529.10\,\mathrm{mL^{-1}}$ on average, translating to a $N_{\mathrm{INPatm}}$ of $0.53\,\mathrm{L^{-1}}$.

From here onwards we will focus mainly on INP characteristics at $T = -25\,^\circ$C. Although, usually only a relatively low percentage of droplets was frozen at this temperature ($17 \pm 20\,\%$), every single sample showed some amount of freezing here. Only one sample was completely frozen prior to reaching $-25\,^\circ$C. Furthermore, we do not expect much influence from background freezing at this temperature (Fig. 2). Figure 6 displays the INP concentrations of the ice core at $-25\,^\circ$C. We chose to show the data both on a linear (a) and logarithmic (b) scale, so the reader can see the typical variation in the INP

concentrations, but is still able to identify samples with higher than usual concentrations more easily. Several important findings arise from the figure. The observed range of variability in the INP concentration is about $1-2$ orders of magnitude. We find somewhat higher and more variable INP concentrations for the more recent samples as compared to the rest of the time series. Yet, there is no obvious trend. Further, back-to-back samples that differ by less than a year (brown, black and green symbols) typically show a comparably low variability, with the exception of two samples from 1475. These two outliers correspond

to samples, which show a peak in the particle number and conductivity signal from the CFA measurements. Currently, it is unclear if this corresponds to a volcanic eruption, which the data seem to suggest. The Laki eruption in 1783, however, did not increase the INP concentrations. Yet, this may have to do with the type of volcanic eruption of Laki, which is categorized as a mostly effusive eruption. In contrast to typical explosive eruptions, during which vast amounts of ash particles are blasted into the atmosphere, effusive mixed eruptions involve alternating mostly liquid lava fountains and flows. Moreover, the location

of Greenland (and the ice core drilling site) situated upwind of the eruption source may contribute to the lack of an increased INP burden over the Greenland ice-sheet in 1783. Cryptotephra from the Laki eruption were only detected at one ice core site (GISP2, Summit; Fiacco et al., 1994). The effusive nature of the eruption and the location away from the main wind direction appear to be reflected by the CFA and IC measurements, which do not find increased particle loads during the Laki eruption, but show the most pronounced peak in conductivity and the $SO_4^{2-}$ concentration. Interestingly, other samples with a peak in

the dust signal did not always translate to high INP concentrations. Overall, however, we did find a moderate yet significant correlation between particles larger $1.2\,\mu$m and the INP concentration over a wide range of temperatures (Tab. 1). This points to a terrestrial source of INPs. The dust signal in Greenlandic ice cores is mainly associated with long range transport from East Asian deserts (Bory et al., 2002; Schüpbach et al., 2018). For certain temperatures we find significant correlations between the INP concentration and $Ca^{2+}$ and the conductivity as well. If the data is grouped into subsets according to Fig. 1 however, we





find that the correlation breaks down for the 10 year samples and the modern day samples, but increases for special events and seasonal samples (Tab. 2). A complete correlation analysis can be found in the supplement (Tab. S1).

## 3.1 Pre-Anthropocene vs. modern era INP concentrations

Figure 7 presents the frozen fraction vs. temperature spectrum of each sample between 1960 and 1990 in a contour plot.
Comparing Fig. 7 to Fig. 5, it is visible at first glance that more droplets froze at warmer temperatures (yellow colors) for the modern day samples than for the 10 year samples, which cover the complete time from 1370 to 1990. We like to point out here that the topmost part of the ice core is made up of relatively porous firn, which is more prone to post-coring contamination of dust as compared to the rest of the ice core. Unfortunately, we cannot entirely exclude the possibility that differences emerged or are enhanced due to post-coring contamination of the firn, as the ice core was stored for some time, despite the CFA
decontamination technique. Furthermore, the results could potentially be intrinsically influenced to some degree by differences in sampling frequency and time coverage. That being said, going forward we will compare the ice nucleation characteristics of these two data sets in more detail (i.e. 31 samples from 1960 to 1990 and 59 samples from 1370 to 1950). The four samples that originally overlapped with both data groups (1960, 1970, 1980 and 1990) will only be included in the modern day subset from now on. We believe this differentiation between pre- and post-1960 samples to be reasonable, as the mid
of the 20th-century has been recently proposed to indicate the beginning of the Anthropocene (Subramanian, 2019). Note, however, that this comparison differs from the commonly used pre-industrial vs. present day distinction mostly for practical reasons. Furthermore, we excluded the sample from 1630 in most of the following analysis because it was the only sample that was completely frozen before $-22\,^\circ$C and would have therefore introduced a bias at colder temperatures, at which it was not possible to specify an INP concentration. Moreover, as stated previously a contamination prior to the INP analysis cannot be
excluded completely for this sample.

Figure 8 illustrates the statistical freezing properties of both data groups using a box-whisker diagram. The figure confirms the previously observed finding that the modern day samples generally show higher frozen fractions at the same temperature. Furthermore, they exhibit a higher variability than the pre-1960 samples for most nucleation temperatures. The differences intensify at a medium supercooled temperature range ($-23\,^\circ$C to $-26\,^\circ$C). On average, the INP concentration of the modern
day samples are 1.85 to 3.35 times higher than the 10 year samples in this specific temperature range (Fig. 9). We tested the significance of these differences with a two-sided T-test. We found that the average INP concentrations of modern day and pre-Anthropocene samples are in fact significantly different from one another at $-23\,^\circ$C (p < 0.0181), $-24\,^\circ$C (p < 0.0008), $-25\,^\circ$C (p < 0.0011), $-26\,^\circ$C (p < 0.0360) and $-28\,^\circ$C (p < 0.0463). Figure 10 compares the relative frequencies of observed INP concentrations of the two groups at $-25\,^\circ$C. Modern day samples follow a relatively broad log-normal probability distribution
with a median ice INP concentration of about $100\,\text{mL}^{-1}$. The INP frequency distribution of pre-1960 samples, on the other hand, is evidently different from the post-1960 samples. Here, we find INP concentrations below $100\,\text{mL}^{-1}$ more frequently. The distribution is more narrow and seems to be right-skewed to some degree, although the log-normal shape is still matched relatively well. Similar results emerge from the INP distributions at $T = -23\,^\circ$C, $-24\,^\circ$C and $-26\,^\circ$C, which can be found in the Supplement (Figs. S2, S3 and S4).





If there was no influence of post-coring contamination, these findings seem to suggest that certain particles that are ice nucleation active in a mid-supercooled temperature regime may be more abundant in today's atmosphere. We already hypothesized about possible candidates in the introduction (e.g. enhanced mineral or soil dust particles due to desertification, land-use change and agriculture, metals from industrial processes and/or fire particles from biomass burning emissions). Potential atmospheric
implications of this finding are discussed in the conclusions.

### 3.2   Chemical composition of single particles

In total 308 particles were analyzed by SEM in the three selected samples (1977: 36 particles, 1680: 188 particles, 1630: 84 particles). Overall, the chemical composition was very similar between the three samples. The dominant part (287 particles – 93%) were soil particles, i.e. alumosilicates and silicon oxides (Fig. S5a). Besides the soil particles, 19 iron-rich particles,
1 titanium-rich and one calcium carbonate particle were found. Most particles are detected in the size interval $1-2\,\mu m$ (only 13 particles above $5\,\mu m$ diameter). Looking at the minor elements in the alumosilicate particles and the typical elemental ratios, most of the detected alumosilicates will be feldspars (and here more sodium and potassium feldspars), amphiboles and pyroxenes. Besides this, some quartz and clay minerals were also found. Unfortunately, due to the low number of analyzed particles, we were unable to determine significant differences in particulate composition of the particles and size distribution
in the three samples.

Moreover, a few fly ash particles (8%) were found on the most recent sample (1977), which indicates a anthropogenic chemical signature (Fig. S5b). Otherwise there was no obvious distinction between the modern-era sample and the other two samples with regards to their chemical composition with the limited analysis presented here.

### 3.3   Seasonal cycle of INPs during a high resolution period

Here we present a case study of 12 samples with an improved time resolution (only $1-2$ months each). These samples represent a period of about 1.5 years in $1463-1464$ (ice core depth: $87.647\,m$ to $87.848\,m$), thus, covering a continuous annual cycle. Figure 11 shows the temperature at which 50% of the droplets were frozen ($T_{50}$). The $T_{50}$ temperature is a simple but meaningful metric, which indicates how ice active a sample is. For the 12 samples $T_{50}$ ranged between $-28.2\,°C$ and $-31.1\,°C$. The entire $T_{50}$ range of the ice core is from $-20.2\,°C$ to $-32.1\,°C$. Thus, with a span of only $2.9\,°C$ the seasonal samples have a
significantly smaller range in $T_{50}$ (76% smaller) than all samples from the ice core. Hence, the long-term INP variation of the ice core is considerably larger than the short-term variation in that specific year.

The number of data points are too few for an in depth seasonal analysis, but the ice nucleation activity of the samples does seem to follow a clear annual cycle. Interestingly, we find a similar annual pattern for various other IC and CFA parameters (lower panel of Fig. 11). The best correlation is found between $T_{50}$ and the log of the concentration of insoluble particles with
spherical diameters larger than $1.2\,\mu m$ (R = 0.87, N = 12, p < 0.0003). The minima and maxima of both parameters are largely the same. In addition, even small fluctuations in $T_{50}$ are reflected in the particle concentration. The results indicate that a higher particle concentration triggers earlier freezing, which is intuitive. A higher number of particles in a droplet means that there is a greater probability of the droplet to contain an INP. These findings suggest that the INP concentration is subject to the annual





dust input in Greenland. Bory et al. (2002) show that the main dust source in Northern Greenland is the Taklamakan Desert in Northern China. At the beginning of the monsoon season, the dust particles are transported to Greenland within a few days via the jet stream and cause the annual maximum dust input for Greenland in spring.

The clear existence of a seasonal cycle and the significant correlation of $T_{50}$ with the particle concentration also shows that the INP background was chosen conservatively and thus confirms the reliability of the data up to this point (at least for the deeper, less porous firn section of the core). Furthermore, the data suggest that even at low temperatures the ice nucleation behavior of the ice core samples is induced by actual atmospheric perturbations.

## 3.4   Comparison with literature data

Hartmann et al. (2019) found no long-term INP trend in their recent study, in which they analyzed ice core samples from
Greenland and Svalbard with two droplet freezing devices. They found the overall range of observed concentrations to be comparable to present day concentrations. INP concentrations did not seem to be influenced by either anthropogenic impacts or volcanic eruptions. Furthermore, sub-year samples showed a large variability, which was as high or even higher than the total range of the other samples from the ice cores. Hartmann et al. (2019) regularly observed an early nucleation onset, which they interpret as an influence from particles with biological origin.

In general, past atmospheric INP concentrations of this ice core study align reasonably well to the lower end of INP concentrations currently observed in the atmosphere (Fig. 12). INP concentrations in this figure comprise data from vastly different environments and range over 4 orders of magnitude at a certain temperature (Kanji et al., 2017). However, our INP concentrations are significantly lower than those presented in Hartmann et al. (2019). Note that the data is difficult to compare as the freezing spectra do not overlap very well. Ice nucleation in this study occurred largely below $-25\,°C$, when most droplets were
already frozen in the study of Hartmann et al. (2019). At $T = -20\,°C$ Hartmann et al. (2019) found atmospheric INP concentrations from about 0.004 to about 2 $L^{-1}$, whereas we found INP concentrations between 0.002 and 0.2 $L^{-1}$. The conversion to atmospheric concentrations was handled differently in both studies, but this does not explain differences up to one order of magnitude. This disparity may arise from experimental, methodological and or geographical differences. Unfortunately, large discrepancies between different INP counters are relatively common, even in controlled laboratory environments (Hiranuma
et al., 2015; DeMott et al., 2018; Hiranuma et al., 2019), and can often not be fully explained. Furthermore, the "short-term" variability of adjacent seasonal, dust and volcano samples were usually lower than the overall variability of the total ice core samples in our study. Whether the INP concentration was influenced by volcanic eruptions or not cannot be assessed conclusively at this point. The Laki eruption of 1783 did not increase the INP concentration, however two samples from 1475 indicate volcanic dust particles with ice nucleation potential.

Similarly, we cannot fully exclude an anthropogenic impact on the Arctic INP population. In fact, some evidence indicates that the concentration of INP active at medium supercooled temperatures has changed significantly after 1960, where we found higher and more variable INP concentrations, compared to the rest of the ice core samples. However, we cannot fully rule out post-coring contamination as the cause for the observed differences. It is possible that this result could not be observed by Hartmann et al. (2019) for several reasons. First, Hartmann et al. (2019) mainly investigated the temperature regime from





−10 °C to −20 °C, while significant differences occurred in our study only below −22 °C. Second, the overall temporal distribution of samples in Hartmann et al. (2019) was less regular, had a coarser time resolution and included only few samples prior to the year 1735. Moreover, we specifically designed our sample selection (regular time intervals of 10 years, increased sample frequency after 1960) with this scientific question in mind.

## 4 Conclusions

Ice nucleating particle concentrations of ice core samples were measured in the immersion freezing mode by the FRIDGE droplet freezing assay. This analysis provides valuable insights into atmospheric variables related to microphysical cloud processes of the past six centuries. A process-based approach was chosen to estimate order-of-magnitude atmospheric concentrations from the directly observed ice concentration. Ice samples were selected for INP analysis following a systematic protocol.

Overall, the samples were not particularly ice nucleation active, with freezing occurring predominantly below −25 °C. However, a group of high resolution samples displayed meaningful and intuitive results – following a seasonal cycle – even at comparably low temperatures, where some influence from background freezing would usually be expected. Furthermore, we found significant correlations between concentrations of INPs and certain aerosol species for a broad range of temperatures. We did not observe a clear trend over time in the INP concentration. Yet, it appears that the population of particles acting as INPs at medium supercooled temperatures has increased after 1960. It cannot be ruled out that the observed results are caused by an enhanced fraction of particles originating from anthropogenic activity. Alternatively, differences may have been caused by post-coring contamination, which is likely more relevant for these samples as they stem from the more porous firn layer.

Still, this prompts the question which atmospheric implications can be expected, if there were more INPs today than in a pre-industrial atmosphere. In general, several mechanisms can be considered by which an increased number of INPs would affect cloud formation processes and consequently radiation interactions (DeMott et al., 2010; Murray, 2017). First, it is thought that the lifetime of mixed-phase clouds is shortened in the presence of ice nucleating particles. Ice crystals will form more rapidly at the expense of cloud droplets and water vapor, leading to earlier precipitation. Since supercooled clouds (e.g. stratus or altocumulus) usually have a net cooling effect, more INPs will consequently decrease this cooling effect due to the shorter lifetime. Similarly, cirrus clouds are also expected to have a shortened lifetime in a high INP scenario. Compared to a cirrus cloud at which cloud droplets predominantly freeze homogeneously in a narrow temperature range, cirrus clouds that formed by heterogeneous ice nucleation will have fewer but larger ice crystals. Since in this case ice crystals have formed earlier, they have had more time to grow and hence will sooner fall out of the cloud. These upper tropospheric clouds typically have a net warming effect. A reduced lifetime will decrease this effect. Furthermore, different proportions of water droplets to ice crystals obviously also alter the cloud albedo. Unfortunately, it is rather unclear which of these effects dominates overall.

The apparent finding presented in this study is not to be generalized easily. It investigates only one location in the Arctic, which is quite isolated from direct human influence. It is also possible that, for example, urban aerosols and/or coatings reduce the ice nucleation properties of the INP population as a whole under more direct anthropogenic influence. We stress




that the observed differences in the average INP concentration at specific temperatures are statistically significant, but the implications are far from certain. We recommend that our measurements should be repeated and verified in other studies, where we recommend to obtain seasonal resolution of the data, as the existence of a clear seasonality represents an effective check for the quality of the INP results.

There is a strong need to investigate INP concentrations of the past, and ice cores provide an unique and feasible opportunity to do so. We suggest to analyze the freezing properties of ice core melt water at different locations worldwide – we feel that other Arctic, Antarctic and mid-latitude Alpine ice cores are equally of interest. Beyond researching a possible anthropogenic INP signal, a multitude of ice core studies could improve our understanding of regional sources and geographical differences of INPs over otherwise inaccessible time scales. Furthermore, we plan to expand our ice core analysis to include a more

rigorous, systematic study analyzing the chemical and morphological composition of insoluble aerosol particles as well as their size distribution by scanning electron microscopy. Moreover, future studies would greatly benefit from more comprehensive and precise knowledge about the present day INP concentration (atmospheric and fresh snow) and its variance, as well as dry and wet deposition metrics at the actual ice core drilling site. Finally, a modeling study could help identify (possibly antropogenically altered) INP source regions and estimate the potential atmospheric impact that could be expected from a

threefold increase of INPs at $-24\,^{\circ}\mathrm{C}$ since the mid of the twentieth century, as it was seen in this study.

*Data availability.* The INP data will be made available using the Data Publisher for Earth & Environmental Science PANGAEA (https://www.pangaea.de/).

*Author contributions.* JS and HB designed the conceptional idea of the presented manuscript. DK, SR and JS performed the INP measure-
ments. JS compiled and analyzed the INP data with support of DK and SR. JS created the figures. Authors affiliated with the Alfred Wegener Institute and the University of Bern were responsible for the drilling, handling and cutting of the ice core, performed IC and CFA measurements and generated the samples for the INP analysis. ME performed the SEM measurements. All authors took part in the discussion of the results. JS wrote the manuscript, receiving valuable input from HB, JC, HF, MH and TE.

*Competing interests.* The authors declare no competing interests.

*Acknowledgements.* The methods and instruments of the ice nucleation experiment of the University of Frankfurt were established through
financial support of Deutsche Forschungsgemeinschaft under Research Cooperations SFB 641 and FOR 1525 (INUIT), which is gratefully acknowledged. Long-term financial support of ice core research at the University of Bern by the Swiss National Science Foundation (SNF) is gratefully acknowledged. Funding for the CFA analyses used in this study was provided by the SNF project iCEP (200020_172506). We thank Remi Dallmayr and Melanie Behrens, who prepared and set up the fraction collection for the the CFA campaign.





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





**Table 1.** Pearson correlation between the INP concentration and selected CFA parameters of the complete data set. Bold coefficients indicate a significant correlation ($p < 0.05$). The number of samples is given in parenthesis.

|  | dust | conductivity | $Ca^{2+}$ |
|---|---|---|---|
| -20 °C | -0.11 (57) | -0.04 (58) | -0.16 (58) |
| -21 °C | -0.06 (91) | -0.03 (92) | -0.10 (91) |
| -22 °C | 0.19 (111) | 0.09 (112) | 0.04 (111) |
| -23 °C | **0.19** (121) | 0.15 (122) | 0.06 (121) |
| -24 °C | **0.26** (125) | **0.20** (126) | 0.12 (125) |
| -25 °C | **0.27** (125) | **0.20** (126) | 0.12 (125) |
| -26 °C | **0.21** (125) | 0.15 (126) | 0.07 (125) |
| -27 °C | **0.24** (124) | 0.11 (125) | 0.09 (124) |
| -28 °C | **0.31** (123) | 0.14 (124) | **0.18** (123) |
| -29 °C | **0.30** (123) | 0.09 (124) | **0.19** (123) |
| -30 °C | **0.34** (120) | 0.04 (121) | **0.20** (120) |
| -31 °C | **0.27** (113) | -0.07 (114) | 0.12 (113) |



**Table 2.** Pearson correlation between the INP concentration and selected CFA parameters of indicated subsets of the data ("events" include the groups dust, volcanic and seasonal). Bold coefficients indicate a significant correlation ($p < 0.05$). The number of samples is given in parenthesis.

| | 10 years | | | modern day | | | events | | |
|---|---|---|---|---|---|---|---|---|---|
| | dust | conductivity | $Ca^{2+}$ | dust | conductivity | $Ca^{2+}$ | dust | conductivity | $Ca^{2+}$ |
| -20 °C | -0.20 (18) | 0.12 (19) | -0.29 (19) | -0.37 (11) | -0.09 (11) | -0.08 (11) | -0.01 (25) | -0.01 (25) | -0.06 (25) |
| -21 °C | -0.13 (37) | 0.04 (38) | -0.19 (38) | -0.25 (17) | -0.35 (17) | -0.44 (16) | **0.35** (34) | **0.44** (34) | 0.29 (34) |
| -22 °C | -0.05 (46) | -0.08 (47) | -0.05 (47) | -0.16 (23) | -0.27 (23) | -0.38 (22) | 0.29 (39) | 0.15 (39) | 0.21 (39) |
| -23 °C | 0.02 (53) | -0.08 (54) | -0.06 (54) | -0.17 (23) | -0.27 (23) | - 0.40 (22) | 0.23 (42) | 0.18 (42) | 0.16 (42) |
| -24 °C | 0.00 (57) | -0.04 (58) | -0.06 (58) | -0.12 (23) | -0.21 (23) | -0.35 (22) | **0.34** (42) | 0.22 (42) | 0.25 (42) |
| -25 °C | 0.01 (57) | -0.08 (58) | -0.11 (58) | -0.09 (23) | -0.21 (23) | -0.37 (22) | **0.35** (42) | 0.22 (42) | 0.26 (42) |
| -26 °C | -0.04 (57) | -0.08 (58) | -0.15 (58) | -0.05 (23) | -0.21 (23) | -0.38 (22) | **0.31** (42) | 0.20 (42) | 0.23 (42) |
| -27 °C | -0.05 (57) | -0.09 (58) | -0.16 (58) | -0.02 (23) | -0.20 (23) | -0.38 (22) | **0.55** (41) | 0.26 (41) | **0.47** (41) |
| -28 °C | 0.05 (56) | -0.10 (57) | -0.03 (57) | 0.01 (23) | -0.18 (23) | -0.34 (22) | **0.50** (41) | 0.23 (41) | **0.46** (41) |
| -29 °C | 0.03 (56) | -0.09 (57) | -0.02 (57) | 0.01 (23) | -0.19 (23) | -0.32 (22) | **0.45** (41) | 0.18 (41) | **0.45** (41) |
| -30 °C | 0.05 (55) | -0.01 (56) | -0.07 (56) | 0.04 (23) | -0.16 (23) | -0.26 (22) | **0.58** (39) | 0.08 (39) | **0.55** (39) |
| -31 °C | 0.14 (53) | -0.17 (54) | -0.07 (54) | -0.01 (22) | -0.21 (22) | -0.31 (21) | **0.38** (36) | -0.06 (36) | **0.39** (36) |



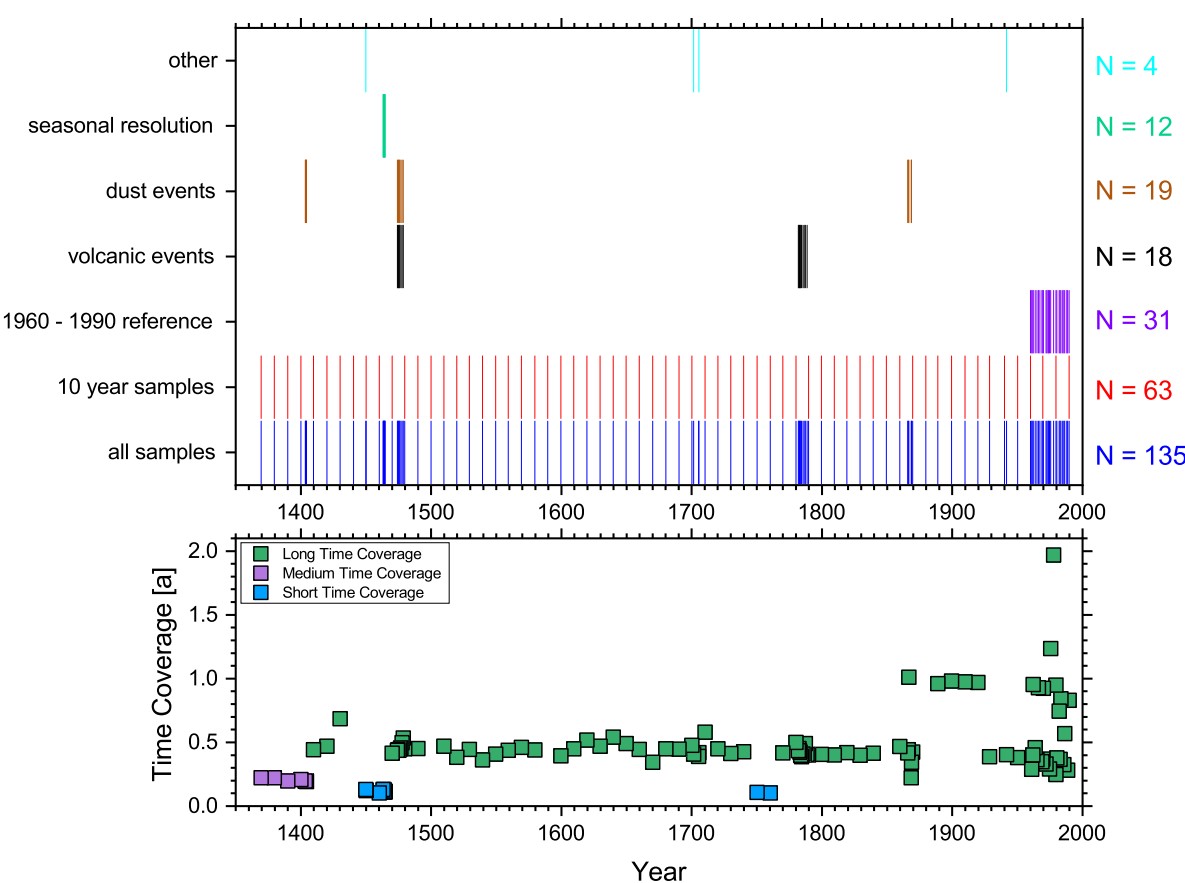

**Figure 1.** a) Temporal distribution and number of ice nucleation samples for different groups (colors). b) Time coverage of ice nucleation samples. The different colors represent a broad grouping into samples averaging over short, medium and longer time periods.

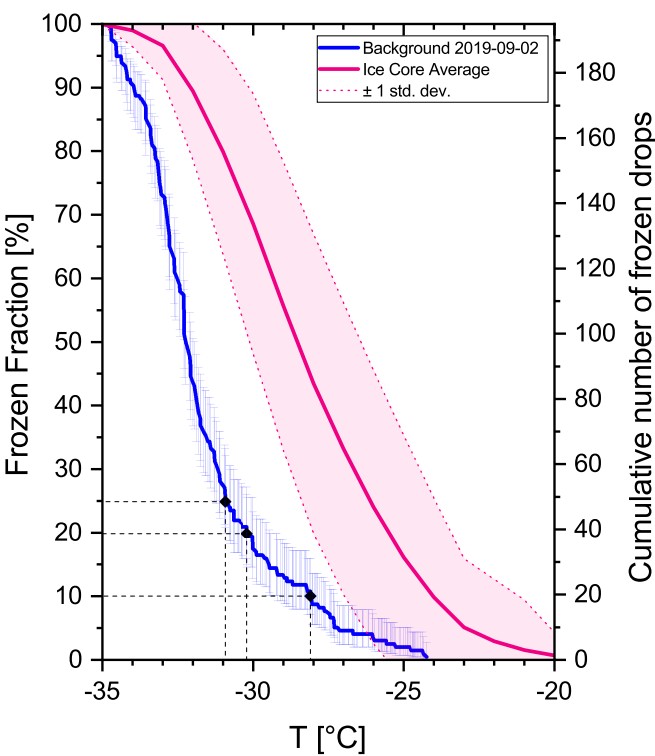

**Figure 2.** Freezing spectrum of a typical background measurement (blue line) compared to the average freezing spectrum from all ice core samples (pink line) ± one standard deviation (pink dotted lines). The background freezing temperatures at the frozen fractions of 10%, 20% and 25% are indicated.





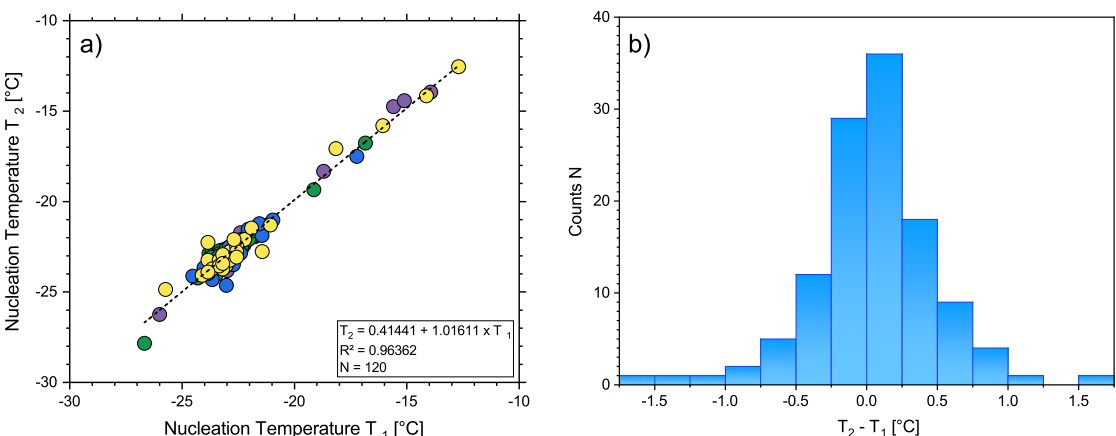

**Figure 3.** a) Freezing temperatures of individual droplets at two subsequent measurements. The different colors correspond to the four experimental runs of 30 droplets each. b) Histogram of the individual droplet temperature difference between the two measurements.

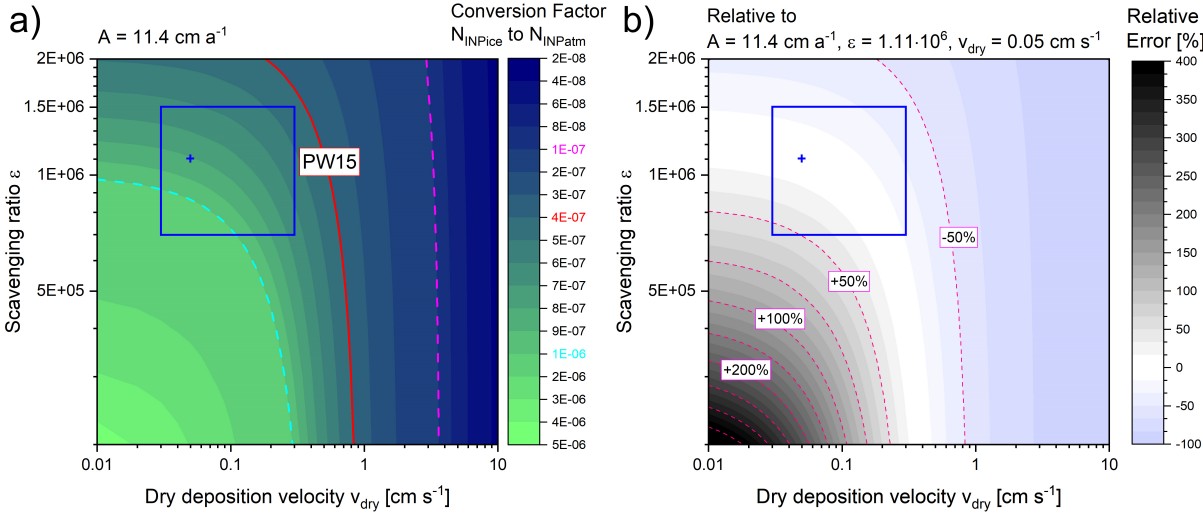

**Figure 4.** a) Conversion factor between the INP concentration per volume of ice and per volume of air depending on dry deposition velocity and scavenging ratio at a fixed accumulation rate. Our best estimate of $8 \cdot 10^{-7}$ is given by a blue cross. A likely range is indicated by a blue rectangle. The conversion factor introduced in Petters and Wright (2015) (PW15, red line) is added for reference. b) Potential errors relative to our best estimate due to uncertainties in $v_{dry}$ and $\varepsilon$.

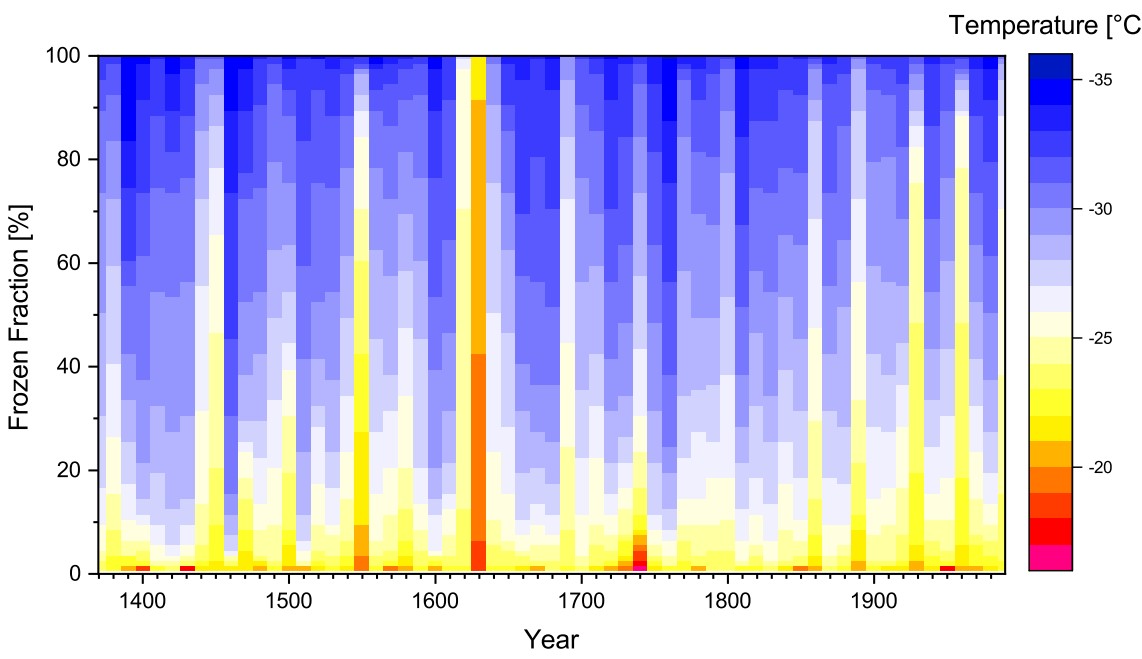

**Figure 5.** Frozen fractions of the samples with regular time intervals of 10 years depending on freezing temperature (colors). Note, that the temporal coverage of an individual sample typically averages over about six month.

**Figure 6.** INP concentrations (right scale: per volume of ice, left scale: per volume of air) at $-25\,^{\circ}$C on both linear (a) and logarithmic (b) scaling. Symbol colors correspond to the different sample groups as introduced in Fig. 1. The ice core's (nonlinear) depth from the top is added for reference on the top x-axis.





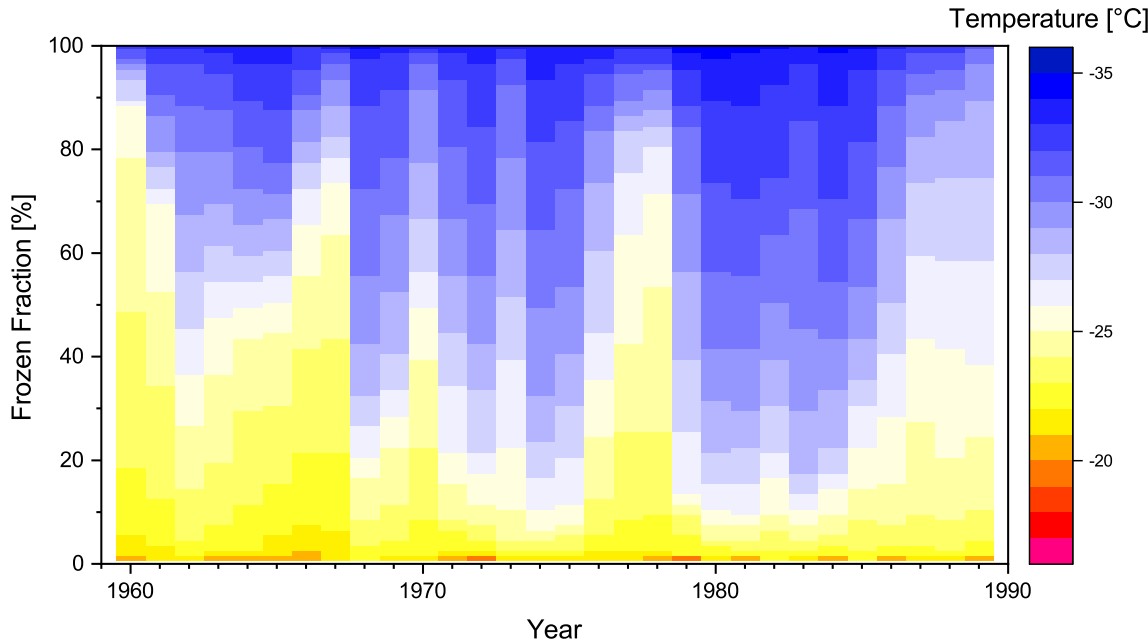

**Figure 7.** Frozen fractions of the modern day samples depending on freezing temperature (colors). Note, that the temporal coverage of an individual sample typically averages over about six month. However, data points are interpolated in time to generate a regular data set.

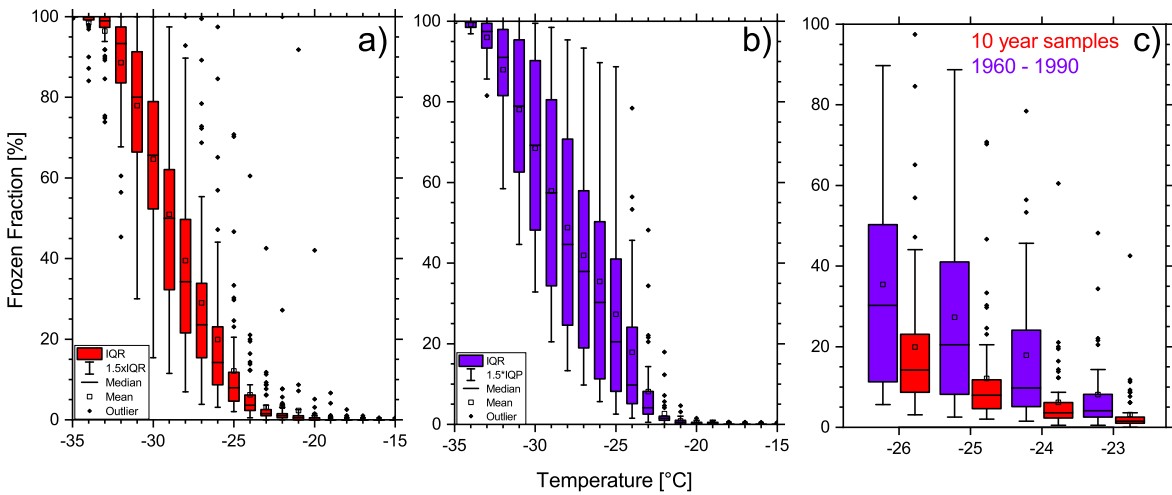

**Figure 8.** Box-Whisker plots of the frozen fraction of the 10 year samples (a) and the modern day samples (b) for the complete temperature range and a detailed comparison of both data sets at medium supercooled temperatures (c).



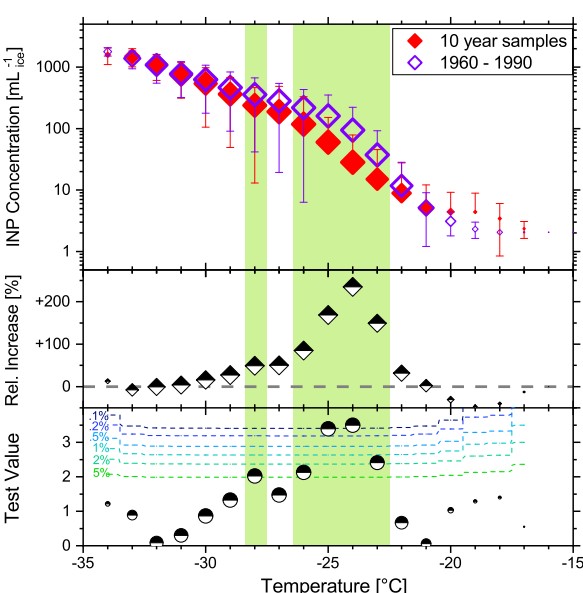

**Figure 9.** Upper panel: Average INP concentrations ± standard deviation (error bars) of the 10 year samples (red) and the modern day samples (purple). Negative error bars are not shown, when the standard deviation is greater than the average. Mid panel: Relative difference in the INP concentration between both groups. Positive values mean modern day samples are higher. Lower panel: Test values of a two-sided T-test to evaluate if the average INP concentrations of both groups are significantly different from each other. Dotted colored lines indicate the significance level of the test. Average INP concentrations that differ significantly from each other at $p < 0.05$ are highlighted in green. Symbol sizes in all panels correspond to the respective number of ice-active samples at each temperature (maximum number for 10 year samples: N = 58, maximum number for 1960 – 1990 samples: N = 31). Note, that the 1630 sample is excluded from this figure.

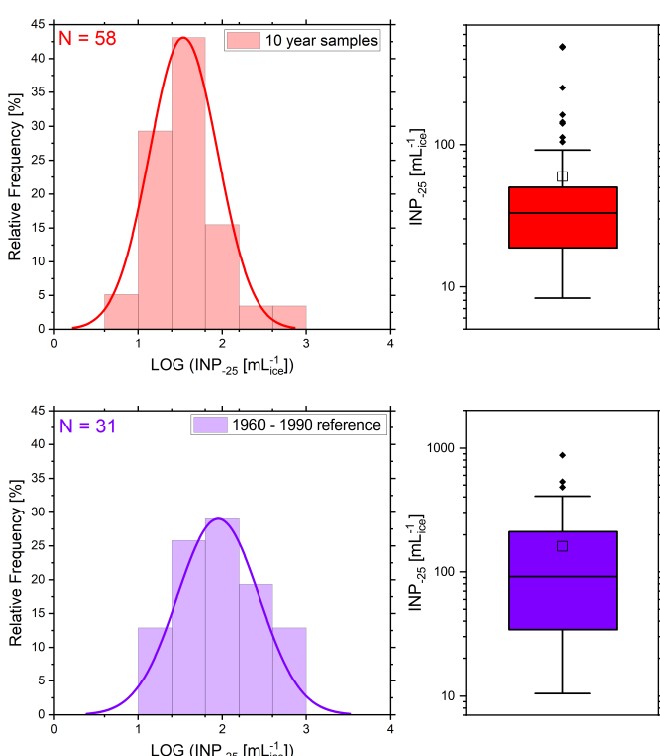

**Figure 10.** Empiric probability density function (bars) of the logarithmic INP concentration at $-25\,^{\circ}\mathrm{C}$ of the 10 year samples (top, red) and the modern day samples (bottom, purple). The data follows a log-normal distribution (fitted curve). The right panel shows the corresponding Box-Whisker plot.





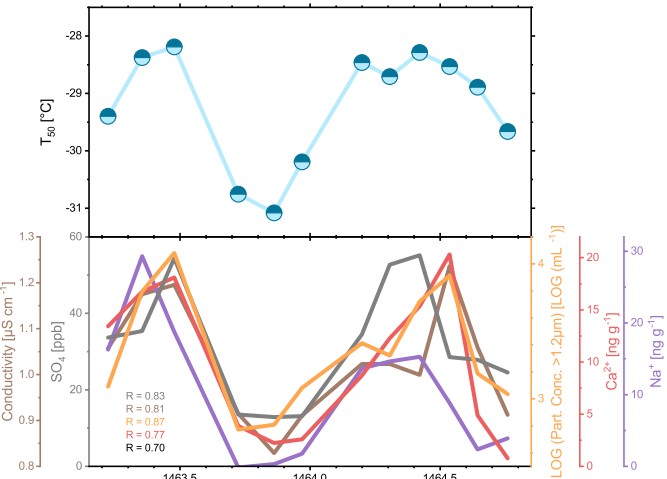

**Figure 11.** Upper Panel: Nucleation temperature at which 50% of the droplets were frozen during a high resolution period around 1464 (sample resolution: $1-2$ months). Lower Panel: Corresponding average concentration of IC ($SO_4^{2-}$) and CFA (conductivity, insoluble particles, $Ca^{2+}$ and $Na^+$) parameters. Pearson coefficients for correlation of parameters to INP concentration are indicated (all $p < 0.05$). Please note that the x-axis may entail some temporal offset, as they refer to individual analyses performed separately on the core which may differ in their depth assignment by $1-2\,cm$.

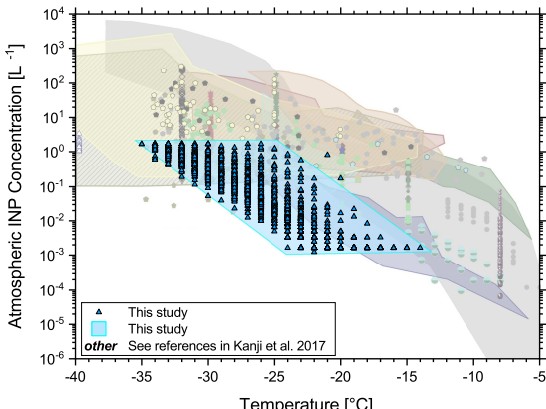

**Figure 12.** Estimated atmospheric INP concentrations of the ice core samples compared to data from other studies of a diverse set of environments as presented in Kanji et al. (2017).