# Peer review of "Ice nucleating particle concentrations of the past: Insights from a 600 year old Greenland ice core"

_Atmospheric Chemistry and Physics, 2020_

## Referee Comment (RC1) · Anonymous Referee #1 · 9 Jul 2020

Review of "Ice nucleating particle concentrations of the past: Insights from a 600 year old Greenland ice core" by Schrod et al.

**General comments:**
This reviewer supports publication of this manuscript in ACP. The research topic - researching INPs in the pristine past conditions - is an important addition to ACP for many reasons; e.g., providing a constraint to climate simulations/projections etc. In spite of many potential artifacts addressed throughout the manuscript, the authors conducted careful and dedicated offline lab experiments, and their findings warrant future follow up studies. Unfortunately, such care was not taken in the preparation of the manuscript (esp. after Sect. 2.2), with the manuscript containing a number of unusual word choices and non-intuitive statements. The reviewer has numerous revisions as listed below. Though most of them are minor, the reviewer would urge the authors of the manuscript to thoroughly proof read their manuscript for improving readability, as this list gets too long.

**Specific and technical comments:**
P1L13: The reviewer suggests the authors to specify dp is in a spherical diameter metric here.

P1L20-21: The reviewer appreciates the authors to be honest scientists extensively addressing some potential artifacts throughout the manuscript. However, the statement of "or some post-corning…" seems unnecessary to conclude the abstract. The reviewer suggests removing this part in the abstract.

P2L6-7: Does the authors mean – "Unfortunately, heterogeneous ice nucleation, which is of primary importance of atmospheric ice formation, has not received…"?

P2L7: As of today → Until now or To date (better word choice)

P2L21: defines → constrains (this seems better fitting here)

P2L27: Although…straightforward, → Evidently,

P2L27: seen → implied

P2L30: The reviewer finds the discussion of anthropogenic INP to be a very important part of the current manuscript and, therefore, wishes that the authors can extend the discussion a bit further? A suggestion for reading is Zhao et al. (2019, Nature Geosci.; https://www.nature.com/articles/s41561-019-0389-4?proof=trueMay) and references therein. Currently, the discussion of anthropogenic INPs is controversial, and the authors can help the community by including an extended discussion here. Doing such may reinforce the paper.

P2L34: Biomass burning aerosol is…least potential contributor to anthropogenic INP.

P3L13: Indeed, soil dust, in part derived from agricultural systems/practices,… Is this what the authors meant? Feel free to modify it.

P3L17: …global land area, of which approx. 9% were identifies…

P3L20-23: Please clarify what "anthropogenic increase in mineral dust concentration" means. Also, a bit more discussion of aerosol particle episodes to Greenland would strengthen the paper.

P3L25-26: write out LINA and INDA? They appear once only, so it seems no abbreviations are necessary.

P3L30 INP analysis is… → Cumulative INP data is presented at temperatures of …

P3L30-:Hartmann et al. (2019) observed… → The authors observed no alternation in the INP concentration over long-term period.

P3L31: Furthermore, → Instead,

P3L32: Please clarify what "dominate the total variability of the complete data set" means to the readers here. One may be able to guess, but the clarification would be appreciated.

P3L32-35: this sentence runs too long. The reviewer suggests separating this sentence into two. For example - … INP concentrations for the last few centuries. Their suggestion was to include…

P4L3: write out FRIDGE.

P4L22: B30. Complementary chemical profiles of…

P4L25: Merge this sentence to the previous paragraph.

P5L2-3: … then split for the online chemical analysis and offline ion chromatography (IC) measurements, where discrete aliquots in vials were used (section 2.4).

P5L4: thus covering → translating to

P5L4: Further, depending on the exact…

P5L5-6: Subsequently, the vials were refrozen and shipped to AWI to measure the concentration of major ions in order to complement the CFA measurements. Keep it simple!

P5L7-8: Some of these samples → Some remained samples

P5L11: The reviewer suggests deleting "Temperature variability ranged…15 hours."

P5L13-14: …were refrozen. (once again is repetitive of re:).

P5L17: longitudinal pertains to vertical sections?

P5L19: The reviewer suggests deleting "absolutely" – the sentence is good and makes sense without this accessory word, so not adding any value to the sentence. Perhaps let the readers decide on their own.

P5L23: purpose-built

P5L24-26: The reviewer suggests deleting "Additionally, trace-elemental…for this manuscript". If the data was not used in this study, then there is no need to report/mention in the reviewer's opinion.

P5L31: are → were

P6L2: are → were

P6L2: The IC provides → In this study, the IC provided

Comment: While the reviewer understands that everyone has their own style on how they use tenses in writing, the use of past/present etc. seems not consistent in this manuscript. The reviewer suggests the authors to improve the consistency on the tenses usage throughout the manuscript. Perhaps, the following site could help the authors:

https://services.unimelb.edu.au/__data/assets/pdf_file/0009/471294/Using_tenses_in_scientific_writing_Update_051112.pdf

P6L6-: We placed a strong emphasis on having a data set with quasi-consistent time intervals for our samples (approximately decadal interval). Furthermore, our sample selection strategy was intended to consider the pre-industrial INP concentration vs. the INP concentration of the recent past (1960-1990).

P6L9:… in the latter time period to rightly match up sub-total sample numbers for each set.

P6L12: …as well as a couple of samples collected before and after it.

P6L12-14: Please clarify what is meant by "Due to…" to the readers. Not intuitive to this reviewer.

P6L15: …were selected. These samples were typically…

P6L23-25: A majority (63%) of the analyzed samples averaged over a time period of 6 ± 2 months. The rest averaged over a shorter (26%) and longer (11%) time. Reads better this way?

P6L28: …aerosol particles are activated to ice crystals by …

P6L30: The reviewer suggests the authors to briefly address the importance of droplet freezing. The question here is that - why was the droplet freezing mode selected and used rather than another? The readers would appreciate a justification.

P7L4-6: are → were (x3)

P7L7: is decreased quickly → was quickly decreased

P7L8: slowly lowered at…until all droplets were frozen.

P7L10: is controlled → was measured (or was it really controlled?)

P7L12: limit → minimize

P7L13-16: Did the authors observe any half-or-less frozen droplets at given $T$s? If so, how did the authors systematically judge the freezing moment/T?

P7L24-31: The reviewer thinks all future tenses should be changed to present.

P7L30: Please provide an overall uncertainties in numeric terms, and discuss these here. The words "substantially" and "higher" seem too abstract.

P8L2: The authors may want to recap the unique importance of 1977, 1680 & 1630 and provide the readers a brief justification of why they were picked for SEM analysis.

Sect. 2.7: Briefly describe the operation conditions of SEM-EDX – beam intensity, WD, SS etc. Were these experimental variables all consistent for all analyses?

P8L11-12: The reviewer suggests excluding "Smaller particles will…". Adding not much value to the section.

P8L16: review the state of the art of → reviewed several

Sect. 2.8: In general, this section can be much more concise. Especially, P8L30-P9L2 seems containing repetitive information and, thus, could be excluded. Three most important sentences in this section are: P9L6 However,…; P9L7 Unfortunately,…; and P9L15 Therefore,… The reviewer suggests the authors to summarize the section by putting simple emphasis on these, and reduce the # of words. The reviewer defers to the point addressed in P9L18-21. No worries. The authors' method sounds.

P10L19-20: This means as well… → This implies that INP concentrations may be higher in ice core samples than ambient INP concentrations at any given time. Or something similar?

P10L26-27: The reviewers agrees about INPs being preserved. The authors may add discussion of Beall et al. (Beall, C. M., Lucero, D., Hill, T. C., DeMott, P. J., Stokes, M. D., and Prather, K. A.: Best practices for precipitation sample storage for offline studies of ice nucleation, Atmos. Meas. Tech. Discuss., https://doi.org/10.5194/amt-2020-183, in review, 2020.).

P10L31-34: Not adding much value to the section. The reviewer suggests removing this part from the manuscript.

P11L3: Where does this 'an order-mag.' come from? Please clarify in the text for the readers.

P13L3: The reviewer accepts the idea of conversion. If the authors are confident it is only +/- 50%, the reviewer suggests massively cut # of texts/words in this section. In general, this section is hard to follow. Spending full 2 pages to derive seems a simple sub-conclusion (i.e., P13L6-7) seems overwhelming. You may list the typical value of each variable (A, v_dry, and epsilon) +/- 'reasonable' upper/lower ranges (that correspond to shape a blue square in Fig. 4) in a table format to reduce # of total words. For that matter, the reviewer wonders if Fig. 4 is really needed and meaningful. A different presentation (again, tabular format) may be considered.

P13L13-15: This paragraph seems not fitting here.

P13L28: very steep freezing → local maximum in – or something similar

P13L29-30: → We verified a reproducibility of our results by confirming two separate measurements agreed each other. This verification eliminated the contamination during our FRIDGE measurements.

 Does this what the authors mean?

P14L4: → they showed a frozen fraction of only 0.7% on average.

P14L7: average in ice concentration --> average $N_{INPice}$

P14L9: Here → At this temperature,

P14L10: From here onward,… → Next, our characterization of INPs at -25°C is specifically discussed.

P14L11-12: …every single sample… → all samples showed some droplet freezing events at this T.

P14L14: , so the reader can see → in order to clarify

P14L15: The reviewer suggests deleting ", but is still…" – not much value added.

P14L16: arise from → can be inferred from

P14L17: delete "somewhat" and specify/clarify what include "more recent samples" in the main text.

P14L22: Yet, → Nevertheless,

P14L30 moderate yet significant → notable

P14L34-P15L2: Delete 'however' and re-write the sentence to clarify what the authors mean to the readers.

P15L6: We like to point out here → It is noteworthy

P15L11: That being said, going forward → Regardless,

P15L12-14: "The four…" – the reviewer could not understand what it meant. Please rephrase and clarify the sentence.

P15L14—16: → The observed difference between pre- and post-1960 samples is based on Subramanian (2019), which defines the 20$^{th}$ century as the beginning of the Anthropocene. Keep it simple, and delete "Note, however,…" – not much adding in.

P15L17-19: But, then, excluding it also biases the authors' data… It is an important outlier, correct? It can be still excluded, but the reviewer suggests the authors to provide a better (and more constructive) justification to exclude it in the text.

P16L1: delete "seem to"

P16L7: Only 36 particles for 1977. Please provide a justification for this small #.

P16L11-13: Please provide reference(s). "will be feldspars" sound awkward. Please rephrase it.

P16L16: How did the authors define "fly ash" through SEM-EDX? Reference(s)?

P16L17: No notable difference found here might be due to limited # of particles analyzed, correct? If so, it should be stated in the text.

P16L27:-28: does seem to follow → shows

P16L33-P17L7: The reviewer suggests the authors to soften the tone regarding the annual cycle. Yes. It is nice to see the seasonal cycle exists in this subset of samples, but the authors might need to be careful on not generalizing it as a bold conclusion here. The authors need to make it clear in the text in this particular section that this applies to only what they have analyzed for. Otherwise, please provide a proper justification why the authors believe the seasonal cycle could persist for other eras.

P17L1: How about an episode of dust along with Atlantic Monsoon? How about Iceland etc.?

Suggested reading:
Iceland is an episodic source of atmospheric ice-nucleating particles relevant for mixed-phase clouds
A. Sanchez-Marroquin
https://advances.sciencemag.org/content/6/26/eaba8137.abstract

P17L30-P18L4: The reviewer appreciates the authors being careful, honest scientists by these statements here and elsewhere in the manuscript. Nonetheless, this part (right before the conclusion!) may give a very negative impression about the authors' study to the readers. Scattered concern statements throughout the manuscript bothers this reviewer, at the least. The authors may compile their concerns here and there regarding all uncertainties in Sect. 2.9 in a brief manner. The readers would understand that the results come with uncertainties, and the authors do not need to be too sensitive to sound.

P17L23: Fig. 11 tells the reviewer that the diversity may derive from the concentration and size of dry & wet deposited particles rather than the listed differences? The variability due to composition is ruled out in Sect. 3.2, correct? Please clarify.

P18L11: particularly → significantly or substantially?

P18L12: group → selected subset

P18L14: recap and specify "certain aerosol species" here for the readers.

P18L20: Delete "several mechanisms can be considered by which". The sentence makes sense without it.

P18L31: The reviewer strongly agrees☺

P19: Perhaps, one of top priorities for the future ice core INP research includes the assessment of particle size distributions in liquid samples by DLS etc. The authors may elaborate it as an outlook? Connecting INP properties to aerosol propensities may resolve some raised concerns?

Tables 1 & 2: Add "Temperature (°C)" as the first column header, and delete °C from the send row.

Table 2: What are "dust, volcanic and seasonal"? Please clarify within the table caption.

Fig. 1 caption: → Time coverage of the samples selected for assessing IN properties.

Fig. 1 caption: longer → long

Fig. 2: Adding the least active spectrum from the core sample (P9L14) may increase the visual importance of this figure.

Fig. 3: The authors may superpose the 1:1 ratio line on top of the fit line. Doing such reinforce the authors' point in a visible manner.

Fig. 6: INP [L^-1_atm] or N_INP_atm? Perhaps, the authors may choose one way to improve the consistency throughout the manuscript.

Fig. 7 caption: "However, data…" – the reviewer did not understand this. Please clarify.

Fig. 7 caption: Delete "Note, that".

The reviewer enjoyed reading it. Hope some of suggestions/comments made here help the authors.

---

## Referee Comment (RC2) · Anonymous Referee #2 · 10 Jul 2020

The authors have made a great effort in trying to reconstruct from the analysis of an ice core the atmospheric concentration of ice nucleating particles (INPs) in the atmosphere over Central Greenland between the years 1370 and 1990. It is only the second such attempt, after a similar but less comprehensive study published last year by another group. Overall, the manuscript is clearly written. Everything is well explained. The text is easy to follow. Data are arranged in a meaningful way in Tables and Figures. Half of the main text is description of methods. Interpretation of results is cautious, if not hesitant. It is here that I see some room for improvement, apart from a few other, minor issues.

[Figure]

The most surprising outcome of this study, from my point of view, is the narrow range of INP concentrations in ice and atmosphere during a period in which Earth has seen a tenfold increase in land area used for agriculture (Pongratz et al., 2008). Ploughing of the North American prairies and the Russian steppes during the past two centuries has greatly accelerated wind erosion with drastic consequences, like the harvest failure of 1891 in the Russian steppes (Moon, 2005) and the Dust Bowl situation in the USA during the 1930s. Also intensive grazing by exploding numbers of domesticated animals has had its share in fostering wind erosion during that time (Neff et al., 2008). Other than desert dust, soil dust from more fertile land carries INPs active at moderate supercooling (O'Sullivan et al., 2014). Therefore, I would have expected to see growing number concentrations of INP active at temperatures at around -15 °C or warmer in samples deposited over the last two centuries. However, this does not seem to be the case. Only 3% of all samples, each consisting of 0.5 mL of melted ice, contained INPs active at -15 °C. There are at least two plausible explanations for this observation. First, it could be that anthropogenically caused dust in the midlatitudes was not transported in detectable quantities to the Arctic and deposited in Central Greenland. The overwhelming majority of dust and INPs deposited in the Arctic probably originates from regions north of 60 °N in America and Eurasia, latitudes not much affected by landuse change in the past. Regions in North America located south of 60 °N probably contribute less than one percent to the total surface dust concentration in the Arctic (Groot Zwaaftink et al., 2016, Table 3 therein). Thus, large-scale landuse change and increased wind erosion of fertile soils in the midlatitudes following the colonisation of North America by settlers mainly from Europe may indeed not have had a marked effect on INPs deposited in Central Greenland, although it clearly increased dust deposition in the midlatitudes (Neff et al., 2008).

Another explanation for landuse change over the past centuries not being reflected in the INP record of the analysed ice core could be a loss of IN-activity, in particular the loss of biological INPs that dominate the spectrum at temperatures warmer -15 °C (Murray et al., 2012). Deactivation might have happened during decades and centuries

in ice or during sample preparation, in particular during melting of the core and while the samples were in liquid form. Since Hartmann et al. (2019) found clearly enhanced INP activity between -5 °C and -15 °C even in some older sample (year 1484), sample preparation may be the more relevant issue. It would be interesting to know the temperature on the hot side of the instrument in which the ice core was melted. Further, for how long, in total, were samples in liquid form between the first melting of the core and INP analysis? Evidence pointing at a partial loss of INPs is in Figure 9 of the manuscript in discussion. It shows from -30 °C to -24 °C increasingly larger INP concentrations in the modern, as compared to the older samples. The relative difference between modern and older samples collapses quickly towards the warmer end of the temperatures scale. I would have expected this difference to continue increasing further until the warmest temperature is reached at which INPs are detectable. Maybe there is no difference at warmer temperatures detectable today because INPs active above -22 °C had lost their activity before INP analysis? In my experience, any challenge put to a population of INPs, such as warming or storage in water, always leads first and foremost to a loss of those INPs that are active at the warmest temperature. The warm temperature "bulge" in a cumulative INP spectrum disappears with increasing severity or duration of a challenge, resulting in the cumulative spectrum approaching a linear shape on a log-scale. The same applies to certain mineral INPs (Harrison et al., 2016, their Figure 4a, top panel). Partial deactivation most likely results in the remaining part of the INP population becoming increasingly homogenous, a guess supported by Figure 8 in the discussed manuscript: the distribution of frozen fractions at a specific temperature was much narrower for the older samples (pre- 1960) as compared to the modern samples (1960 to 1990). The majority of fragile INPs, which may have been present at the time of deposition, and still are to some extent in the samples from 1960 onwards, may have been lost, leaving behind a relatively homogenous population of very stable INPs. To summarise, very limited dust transport from the midlatitudes, where most landuse change has happened in past centuries, and deactivation of INPs active at temperatures warmer than -20 °C may explain why the concentration of INPs

in the ice core is confined to a narrow range and does not reflect the growing human impact on land over the past few centuries. These considerations are of cause speculative, but I hope they encourage the authors to push their interpretation a bit further.

Minor issues

Page 9, lines 19-20: I am always at a loss when told that results "...should be interpreted with care." Is not every interpretation or conclusion based on empirical evidence a preliminary one and absolutely true statements only to be found within closed systems (mathematics, logic)?

Page 9, line 32: Why use the number of frozen droplets and not the number of INPs in the assay (INPs in 195 droplets) as the criterion from which to estimate uncertainty?

Page 15, line 26: The data has a lognormal distribution. Was it log-transformed before the t-test?

Conclusions section: Effects of INPs on cloud radiative properties are mentioned and I wonder whether the very small number concentrations found in the ice core, and the difference between 1960 to 1990 or before, are indeed in a range where they might lead to differences in radiative properties?

Regional sources and geographical differences in INPs may not only be accessible through the analyses of ice cores but also through modelling approaches making use of historical records of land cover.

Figure 1b: Would it be possible to indicate the season for samples with time coverage below one year?

Figure 2: I would like to see more than one background measurement.

References

Groot Zwaaftink et al., Substantial contribution of northern high-latitude sources to mineral dust in the Arctic, Journal of Geophysical Research: Atmospheres,

doi:10.1002/2016JD025482.

Harrison et al., 2016, Not all feldspars are equal: a survey of ice nucleating properties across the feldspar group of minerals, Atmospheric Chemistry and Physics, doi:10.5194/acp-16-10927-2016.

Hartmann et al., 2019, Variation of Ice Nucleating Particles in the European Arctic Over the Last Centuries, Geophysical research Letters, https://doi.org/10.1029/2019GL082311.

Moon, 2005, The environmental history of the Russian Steppes: Vasilii Dokuchaev and the harvest faillure of 1891, Transactions of the Royal Historical Society, https://www.jstor.org/stable/3679366.

Murray et al., 2012, Ice nucleation by particles immersed in supercooled cloud droplets, Chem Soc Rev, doi: 10.1039/c2cs35200a.

Neff et al., Increasing eolian dust deposition in the western United States linked to human activity, Nature Geoscience, doi:10.1038/ngeo133.

O'Sullivan et al., 2014, Ice nucleation by fertile soil dusts: relative importance of mineral and biogenic components, Atmospheric Chemistry and Physics, https://www.atmos-chem-phys.net/14/1853/2014/.

Pongratz et al., 2008, A reconstruction of global agricultural areas and land cover for the last millennium, Global Biogeochemical Cycles, doi:10.1029/2007GB003153.
* * *

---

## Referee Comment (RC3) · Anonymous Referee #3 · 14 Jul 2020

Review of Ice nucleating particle concentrations of the past: Insights from a 600 year old Greenland ice core

In this study Schrod et al, present the ice nucleating particle (INP) concentrations from a Greenland ice core spanning the past 600 years. The collected data set shows that the concentration of INPs has been rather consistent over the past 600 years. However, since 1960, the concentration and variability in INPs has increased. This has led the authors to suggest that human activities may be influencing INP concentrations, which could have significant impacts on future cloud radiative forcing. I appreciate that the authors are very careful in not over interpreting their results and are very thorough

in addressing potential issues with conversions and contamination. I support the publication of this manuscript and provide some minor technical revisions. Additionally, I think it would be very interesting to extend the analysis to investigate the role of changing atmospheric circulation and rising arctic temperatures may have on the observed changes in INP concentration in this ice core sample.

General comments:

Although all layers of the ice core were treated the same and likely experienced similar temperature variabilities while accumulating on the ice sheet, it would be worthwhile to mention the recently found impacts of the storage on INPs relative to freshly collected samples. For example see Beall et al., (2020) and Stopelli et al., (2014). As the long term storage of the INPs in the ice may contribute to the observed difference between the ice core samples and precipitation samples shown in (Petters and Wright, 2015).

As each of the samples used to probe the concentration of INPs every 10 years only covers a period of 6 months, is the 6 month period roughly the same for each of the 10 yr samples? Based on Fig. 6, the variability over a year (monthly sample from 1463-64) looks to be about an order of magnitude. Therefore, if the 6 months covered by a 10 yr sample differs, some of the variability between the 10 yr samples, albeit a small amount, could be explained.

The same question is also relevant for the modern day samples (Fig. 7) where there are some years with higher activity than others. It would be important to know if the yearly samples (actually only 6 months) cover the same 6 month period for each year.

Here it is shown that the Anthropocene samples are significantly different from the pre-industrial samples. This is a very interesting finding and something that the authors suggest may be due to a change in the dust due to desertification, and other anthropogenic related aerosols that reach the Greenland ice sheet. Although these seem like possibilities, it would be interesting to discuss the potential influence from changes in atmospheric circulation patterns such as the NAO (Pinto and Raible, 2012).

Additionally, it has been shown that precipitation effectively removes precipitation (Stopelli et al., 2015) and as the ice core site is at a high altitude arctic site, it may be extremely sensitive to the temperature and amount of precipitation that falls (removal of INPs) upstream of the site. The fact that an overall increase in IN activity has been observed in more recent, warmer years may be consistent with warmer air masses precipitating over the ice sheet where fewer INPs have been removed upstream compared to previous (colder) years. Therefore, it may be worthwhile to compare the INP concentrations with the reconstructed temperature record over the same period from the ice core.

Minor comments:

Page 3, line 3: Consider adding the following references: Grawe et al., (2016, 2018); Kanji et al., (2020); Ullrich et al., (2016)

Page 3 line 14: Consider adding the following references: Hill et al., (2016); Steinke et al., (2016)

Page 3 line 20: It is highlighted here that the dominant dust sources in Greenland ice cores come from Chinese deserts and the Taklamakan. Therefore, it would be might worthwhile to discuss the observed ability of these mineral dusts to act as INPs. Do they match in terms of INA with the observed INPs found in the ice cores (it seems like they do)? Consider mentioning previous studies on INPs from this region such as Boose et al., (2016); Field et al., (2006); Paramonov et al., (2018); Ullrich et al., (2016).

Page 6 line 14: change "must" to "does"

Page 6 line 20-21: why was the seasonal variability explored in the 1463? Is there a reason for choosing this period? Wouldn't a more recent year make it easier to identify the months of the year as the ice is less compact?

Page 6 line 28: "hast" should be "has"

Page 7 line 5: consider rephrasing "picked up" to "pipetted"

[Figure]

Page 7 line 7-8: Why is FRIDGE kept at 14 C initially? Based on what was stated earlier, the samples were defrosted at 6 C, so why wasn't FRDIGE set to 6 C to minimize the temperature range a sample was exposed to. Granted, all of the samples experienced the same treatment so this likely has no impact on the overall comparison between samples.

Page 7 line 8-10: Do you mean that the Lauda cryostat was used to dissipate heat from the Peltier element. Please rephrase this sentence to make that clearer.

Page 7 line 11: Does the synthetic air flush change the size of the droplets during the experiment via evaporation? If yes, would this be significant enough to increase the concentration of solutes in a droplet such that it may lead to a freezing point depression in the samples? In theory, the colder the cell gets (the longer the experiment lasts) the more concentrated these solutes would become.

Page 7 line 18: Here you mention mL of meltwater but then use mLice when reporting INP concentrations. Consider making the terminology consistent.

Page 8 line 6: Why was the SEM analysis conducted on the samples after being filtered (400 nm pore size) when the highest correlation between INP concentration and particles concentrations was for particles larger than 1.2 microns? Do these large particles make it through the filter?

Page 8 line 27: Check if "microscopical" should be "microscopic" in this case.

Page 9 line 29: Here it is mentioned that the freezing and melting of the same droplets does not influence the ice nucleating ability of the samples. As previously mentioned in the general comments, it might be worthwhile to mention other studies where it was shown that over longer periods, the storage and repeated melting and freezing of samples influenced the ice nucleating ability of samples.

Page 12 line 22: Remove extra "/" after gprecip in first term of equation

Page 14 line 8: please specify that this is the concentration at -20 C as mention of -20

C comes two sentences earlier.

Page 16 line 10-11: How do these large particles make it through the 400 nm pore sized filters described in the methods?

Page 16 line 27: Here it is mentioned that there is a seasonal cycle in INP and although the variability is significantly less than the over the entire period of the study, it may be worth mentioning if the 6 month samples are taken to over the same 6 months in every time point (as said in the general comments).

Page 17 line 15-25: Could some of the differences in the INP concentrations be due to the droplet size used in the studies? Perhaps the small droplet volume in this study makes the measurement of rarer INPs less quantifiable. Additionally, could location differences between sampling sites, lead to differences in the number and efficiency of INPs removed upstream of the sites (Stopelli et al., 2015). For example, Svalbard often experiences periods of relatively warm air masses laden with INPs that would precipitate out before reaching the high altitude location of this core. These points, although briefly mentioned, could be expanded on.

References:

Beall, C. M., Lucero, D., Hill, T. C., DeMott, P. J., Stokes, M. D. and Prather, K. A.: Best practices for precipitation sample storage for offline studies of ice nucleation, Atmospheric Meas. Tech. Discuss., 1–20, doi:https://doi.org/10.5194/amt-2020-183, 2020.

Boose, Y., Welti, A., Atkinson, J., Ramelli, F., Danielczok, A., Bingemer, H. G., Plötze, M., Sierau, B., Kanji, Z. A. and Lohmann, U.: Heterogeneous ice nucleation on dust particles sourced from nine deserts worldwide – Part 1: Immersion freezing, Atmospheric Chem. Phys., 16(23), 15075–15095, doi:https://doi.org/10.5194/acp-16-15075-2016, 2016.

Field, P. R., Möhler, O., Connolly, P., Krämer, M., Cotton, R., Heymsfield, A. J., Saathoff, H. and Schnaiter, M.: Some ice nucleation characteristics of Asian and Saharan desert

dust, Atmos Chem Phys, 6(10), 2991–3006, doi:10.5194/acp-6-2991-2006, 2006.

Grawe, S., Augustin-Bauditz, S., Hartmann, S., Hellner, L., Pettersson, J. B. C., Prager, A., Stratmann, F. and Wex, H.: The immersion freezing behavior of ash particles from wood and brown coal burning, Atmospheric Chem. Phys., 16(21), 13911–13928, doi:10.5194/acp-16-13911-2016, 2016.

Grawe, S., Augustin-Bauditz, S., Clemen, H.-C., Ebert, M., Eriksen Hammer, S., Lubitz, J., Reicher, N., Rudich, Y., Schneider, J., Staacke, R., Stratmann, F., Welti, A. and Wex, H.: Coal fly ash: linking immersion freezing behavior and physicochemical particle properties, Atmospheric Chem. Phys., 18(19), 13903–13923, doi:https://doi.org/10.5194/acp-18-13903-2018, 2018.

Hill, T. C. J., DeMott, P. J., Tobo, Y., Fröhlich-Nowoisky, J., Moffett, B. F., Franc, G. D. and Kreidenweis, S. M.: Sources of organic ice nucleating particles in soils, Atmospheric Chem. Phys., 16(11), 7195–7211, doi:10.5194/acp-16-7195-2016, 2016.

Kanji, Z. A., Welti, A., Corbin, J. C. and Mensah, A. A.: Black Carbon Particles Do Not Matter for Immersion Mode Ice Nucleation, Geophys. Res. Lett., 47(11), e2019GL086764, doi:10.1029/2019GL086764, 2020.

Paramonov, M., David, R. O., Kretzschmar, R. and Kanji, Z. A.: A laboratory investigation of the ice nucleation efficiency of three types of mineral and soil dust, Atmospheric Chem. Phys., 18(22), 16515–16536, doi:https://doi.org/10.5194/acp-18-16515-2018, 2018.

Petters, M. D. and Wright, T. P.: Revisiting ice nucleation from precipitation samples, Geophys. Res. Lett., 42(20), 8758–8766, doi:10.1002/2015GL065733, 2015.

Pinto, J. G. and Raible, C. C.: Past and recent changes in the North Atlantic oscillation, WIREs Clim. Change, 3(1), 79–90, doi:10.1002/wcc.150, 2012.

Steinke, I., Funk, R., Busse, J., Iturri, A., Kirchen, S., Leue, M., Möhler, O., Schwartz, T., Schnaiter, M., Sierau, B., Toprak, E., Ullrich, R., Ulrich, A., Hoose, C. and Leisner, T.: Ice nucleation activity of agricultural soil dust aerosols from Mongolia, Argentina, and Germany, J. Geophys. Res. Atmospheres, 121(22), 13,559-13,576, doi:10.1002/2016JD025160, 2016.

Stopelli, E., Conen, F., Zimmermann, L., Alewell, C. and Morris, C. E.: Freezing nucleation apparatus puts new slant on study of biological ice nucleators in precipitation, Atmospheric Meas. Tech., 7(1), 129–134, doi:10.5194/amt-7-129-2014, 2014.

Stopelli, E., Conen, F., Morris, C. E., Herrmann, E., Bukowiecki, N. and Alewell, C.: Ice nucleation active particles are efficiently removed by precipitating clouds, Sci. Rep., 5, 16433, doi:10.1038/srep16433, 2015.

Ullrich, R., Hoose, C., Möhler, O., Niemand, M., Wagner, R., Höhler, K., Hiranuma, N., Saathoff, H. and Leisner, T.: A New Ice Nucleation Active Site Parameterization for Desert Dust and Soot, J. Atmospheric Sci., 74(3), 699–717, doi:10.1175/JAS-D-16-0074.1, 2016.
* * *

---

## Author Comment (AC1) · 9 Sep 2020

**Response to Anonymous Referee #1**

First of all, we thank the referee for submitting their helpful and productive annotations, which lead to improvements and clarifications within the manuscript.

We prepared a revised manuscript that addresses the questions and comments of the referees. Furthermore, below we explicitly respond to each of the items raised in the comments of anonymous referee #1. These comments are indicated in *italics,* whereas the author's response is presented in blue. Changes in the manuscript are given in green; changes to the supplement are given in purple. A response with "Okay." means we accepted the reviewers' suggestion and implemented it in the manuscript. The differences are also highlighted in separate PDFs using latexdiff. All line and page numbers refer to the ACPD manuscript version, not the revised manuscript.
* * *
*Review of "Ice nucleating particle concentrations of the past: Insights from a 600 year old Greenland ice core" by Schrod et al.*

*General comments:*

*This reviewer supports publication of this manuscript in ACP. The research topic - researching INPs in the pristine past conditions - is an important addition to ACP for many reasons; e.g., providing a constraint to climate simulations/projections etc. In spite of many potential artifacts addressed throughout the manuscript, the authors conducted careful and dedicated offline lab experiments, and their findings warrant future follow up studies. Unfortunately, such care was not taken in the preparation of the manuscript (esp. after Sect. 2.2), with the manuscript containing a number of unusual word choices and non-intuitive statements. The reviewer has numerous revisions as listed below. Though most of them are minor, the reviewer would urge the authors of the manuscript to thoroughly proof read their manuscript for improving readability, as this list gets too long.*

> We thank the reviewer for their careful reading of the manuscript. The long list of language edits, additional ideas and suggested literature are greatly appreciated. We agree that the suggested changes will improve the readability of the manuscript. We will go through the comments listed below one by one.

*Specific and technical comments:*

- *P1L13: The reviewer suggests the authors to specify dp is in a spherical diameter metric here.*

  > Okay.

- *P1L20-21: The reviewer appreciates the authors to be honest scientists extensively addressing some potential artifacts throughout the manuscript. However, the statement*

of *"or some post-corning..."* seems unnecessary to conclude the abstract. The reviewer suggests removing this part in the abstract.

Okay.

- *P2L6-7: Does the authors mean – "Unfortunately, heterogeneous ice nucleation, which is of primary importance of atmospheric ice formation, has not received..."?*

The sentence was phrased this way to illustrate that although ice nucleation is very relevant to precipitation processes in the atmosphere and by extension to snow accumulation in the Arctic, ironically ice nucleation experiments on ice cores have not been sought out by researchers frequently. We rephrased the sentence to make this more clear:

"Unfortunately, heterogeneous ice nucleation, which is of primary importance to atmospheric ice formation and therefore very relevant to Polar snow accumulation, has not received much attention in ice core sciences."

- *P2L7: As of today → Until now or To date (better word choice)*

Okay.

- *P2L21: defines → constrains (this seems better fitting here)*

Okay.

- *P2L27: Although...straightforward, → Evidently,*

Okay.

- *P2L27: seen → implied*

Okay.

- *P2L30: The reviewer finds the discussion of anthropogenic INP to be a very important part of the current manuscript and, therefore, wishes that the authors can extend the discussion a bit further? A suggestion for reading is Zhao et al. (2019, Nature Geosci.; https://www.nature.com/articles/s41561-019-0389-4?proof=trueMay) and references therein. Currently, the discussion of anthropogenic INPs is controversial, and the authors can help the community by including an extended discussion here. Doing such may reinforce the paper.*

We agree with the referee about the importance of discussing the relevance of anthropogenic INPs for the manuscript. We thank the referee for pointing the suggested reading out to us. We added additional text to the manuscript to emphasize the discussion:

"[...] Yet, the significance of anthropogenic pollution particles to atmospheric ice nucleation is still in question. Recently, Zhao et al. (2019) investigated the effects of pollution aerosol to the ice phase in moderate and strong convective systems in a top-down approach using a combination of satellite observations and model simulations. They present evidence that in the moderate convection case, where heterogeneous ice nucleation is more relevant, the ice particle effective radius is increased, indicating that continental pollution aerosol may in fact contain a considerable fraction of INPs. On the other hand, further experimental studies suggest that most anthropogenic aerosol particles are typically poor INPs. For example, Chen et al. (2018) found that the heavy air

pollution of Beijing did not affect the INP concentration in this urban setting in the investigated temperature range from -6 °C to -25 °C. Overall, there are still few studies available on the ice nucleation efficiency of anthropogenic aerosol and some of the presented evidence is conflicting. Although pure pollution aerosols are considered rather inactive INPs, this does not per se mean that the INP population as a whole has not changed at all over the last centuries. On the contrary, it seems rather likely that certain particles with ice nucleating potential may in fact be more abundant in today's atmosphere. [...]"

- *P2L34: Biomass burning aerosol is...least potential contributor to anthropogenic INP.*
   Okay.

- *P3L13: Indeed, soil dust, in part derived from agricultural systems/practices,... Is this what the authors meant? Feel free to modify it.*
   Yes, that's what we meant. Okay.

- *P3L17: ...global land area, of which approx. 9% were identifies...*
   Okay.

- *P3L20-23: Please clarify what "anthropogenic increase in mineral dust concentration" means. Also, a bit more discussion of aerosol particle episodes to Greenland would strengthen the paper.*
   There are several instances throughout the manuscript, where the dust transport patterns to Greenland are discussed. Therefore, we chose not to overly go into details in the introduction. We rephrased the sentence:
   "[...] however a recent increase in mineral dust concentration from these areas due to anthropogenic impacts is not documented."

- *P3L25-26: write out LINA and INDA? They appear once only, so it seems no abbreviations are necessary.*
   Okay. The line now reads:
   "[...] Leipzig Ice Nucleation Array (LINA, 90 x 1 µL) and Ice Nucleation Droplet Array (INDA, 96 x 50 µL)."

- *P3L30 INP analysis is... →  Cumulative INP data is presented at temperatures of ...*
   Okay.

- *P3L30-31: Hartmann et al. (2019) observed... → The authors observed no alternation in the INP concentration over long-term period.*
   We chose to keep our phrasing, as it is more concise.

- *P3L31: Furthermore, → Instead,*
   We chose to keep our phrasing. "Instead" would indicate an opposite finding, but we feel the high short-term variability is an independent finding from the non-existing long-term trends.

- *P3L32: Please clarify what "dominate the total variability of the complete data set" means to the readers here. One may be able to guess, but the clarification would be appreciated.*

  We rephrased the sentence:
  "Furthermore, they found the "short-term" variability of INP concentrations from adjacent sub-year samples to be as large as or even larger than the total variability of the complete data set."

- *P3L32-35: this sentence runs too long. The reviewer suggests separating this sentence into two. For example - ... INP concentrations for the last few centuries. Their suggestion was to include...*

  Okay.

- *P4L3: write out FRIDGE.*

  Okay.

- *P4L22: B30. Complementary chemical profiles of...*

  Okay.

- *P4L25: Merge this sentence to the previous paragraph.*

  We are not quite sure what the reviewer meant by the comment.
  We deleted the line break.

- *P5L2-3: ... then split for the online chemical analysis and offline ion chromatography (IC) measurements, where discrete aliquots in vials were used (section 2.4).*

  We chose to keep our phrasing. In the suggested sentences the reader might think that IC measurements were performed right away, when in fact they were performed sometime later.

- *P5L4: thus covering → translating to*

  Okay.

- *P5L4: Further, depending on the exact...*

  Okay.

- *P5L5-6: Subsequently, the vials were refrozen and shipped to AWI to measure the concentration of major ions in order to complement the CFA measurements. Keep it simple!*

  Okay.

- *P5L7-8: Some of these samples → Some remained samples*

  We chose to keep our phrasing.

- *P5L11: The reviewer suggests deleting "Temperature variability ranged...15 hours."*

  Okay.

- *P5L13-14: ...were refrozen. (once again is repetitive of re:).*

  Okay.

- *P5L17: longitudinal pertains to vertical sections?*
  Yes.
  We deleted "longitudinal" for clarity.

- *P5L19: The reviewer suggests deleting "absolutely" – the sentence is good and makes sense without this accessory word, so not adding any value to the sentence. Perhaps let the readers decide on their own.*
  Okay.

- *P5L23: purpose-built*
  Okay.

- *P5L24-26: The reviewer suggests deleting "Additionally, trace-elemental...for this manuscript". If the data was not used in this study, then there is no need to report/mention in the reviewer's opinion.*
  Okay.

- *P5L31: are → were*
  Okay.

- *P6L2: are → were*
  Okay.

- *P6L2: The IC provides → In this study, the IC provided*
  Okay.

- *Comment: While the reviewer understands that everyone has their own style on how they use tenses in writing, the use of past/present etc. seems not consistent in this manuscript. The reviewer suggests the authors to improve the consistency on the tenses usage throughout the manuscript. Perhaps, the following site could help the authors: https://services.unimelb.edu.au/__data/assets/pdf_file/0009/471294/Using_tenses_in_scientific_writing_Update_051112.pdf*
  We thank the reviewer for noticing the inconsistencies in tense usage. We will carefully proof-read the manuscript again in this regard.

- *P6L6-: We placed a strong emphasis on having a data set with quasi-consistent time intervals for our samples (approximately decadal interval). Furthermore, our sample selection strategy was intended to consider the pre-industrial INP concentration vs. the INP concentration of the recent past (1960-1990).*
  Okay. "We considered these time intervals to be both meaningful and feasible." was added in between the suggested sentences.

- *P6L9:... in the latter time period to rightly match up sub-total sample numbers for each set.*
  We added the following clause to the sentence:
  "[...] in the latter time period to potentially enhance the statistical significance."

- *P6L12: ...as well as a couple of samples collected before and after it.*
  Okay.

- *P6L12-14: Please clarify what is meant by "Due to..." to the readers. Not intuitive to this reviewer.*

  We selected a subgroup of samples to be analyzed in FRIDGE according to the high-resolution CFA data. Some samples were selected, because there was a peak in the high-resolution dust or conductivity signal. However, the discrete INP samples are random means over several months around this peak, which means that a sample with a CFA peak does not necessarily have to have a high mean value. We will rephrase the sentence:

  "Due to the episodic nature of such an event and the fact that the INP samples were automatically collected as multi-month means, the sample containing the high-resolution peak signal does not necessarily need to have an extraordinarily high average value itself."

- *P6L15: ...were selected. These samples were typically...*

  We rephrase the sentence to:

  "Similarly, peak samples in the high-resolution signal of conductivity were selected. Large peaks in the electrolytic conductivity record are most often derived from high sulfuric acid deposition in the ice after volcanic eruptions."

- *P6L23-25: A majority (63%) of the analyzed samples averaged over a time period of 6 ± 2 months. The rest averaged over a shorter (26%) and longer (11%) time. Reads better this way?*

  We rephrase the sentence to:

  "The majority (63%) of the analyzed samples averaged over a time period of 6 ± 2 months. About a quarter of the samples (26%) averaged over a shorter time and 11% over a longer time."

- *P6L28: ...aerosol particles are activated to ice crystals by ...*

  Okay.

- *P6L30: The reviewer suggests the authors to briefly address the importance of droplet freezing. The question here is that - why was the droplet freezing mode selected and used rather than another? The readers would appreciate a justification.*

  We thank the reviewer for pointing this question out. First and foremost, immersion freezing is considered to be the most atmospherically relevant mechanism in heterogeneous ice nucleation for mixed-phase clouds (e.g Murray et al. (2012)). Moreover, using a droplet freezing assay (DFA) feels like the natural choice to study the ice nucleation ability of particles that are already immersed within ice core meltwater, especially considering that a DFA needs only few microliters of water. All other methods would require additional steps of particle generation (e.g. atomizer), which may introduce further contamination sources and would likely require more sample water. We added a few sentences to the manuscript:

  "We focused on the droplet freezing assay (DFA), because 1) immersion freezing is considered to be the most atmospherically relevant process in heterogeneous ice nucleation for mixed-phase clouds (e.g. Murray et al. 2012), 2) the use of a DFA seems to be the natural choice considering that the aerosol particles are already immersed within the ice core meltwater, 3) the technique requires only

a few mL of sample water, and 4) other methods would likely introduce further contamination sources through the particle generation setup (e.g. atomizer)."

- *P7L4-6: are → were (x3)*
  Okay.

- *P7L7: is decreased quickly → was quickly decreased*
  Okay.

- *P7L8: slowly lowered at...until all droplets were frozen.*
  Okay.

- *P7L10: is controlled → was measured (or was it really controlled?)*
  Okay.

- *P7L12: limit → minimize*
  Okay.

- *P7L13-16: Did the authors observe any half-or-less frozen droplets at given Ts? If so, how did the authors systematically judge the freezing moment/T?*
  The moment of freezing was registered automatically by the LabView software as freezing causes a significant change in brightness. However, sometimes the freezing of a droplet begins just as the images is saved. In these cases the software sometimes misses it, and would count the frozen droplet one image later. But we checked every image manually to account for this. However, it is possible that a droplet froze between two measurement images (which are 10 seconds apart). Therefore, the freezing temperature has an uncertainty of 1/6 °C at the freezing rate of 1 °C/min due to this effect.

- *P7L24-31: The reviewer thinks all future tenses should be changed to present.*
  Okay.

- *P7L30: Please provide an overall uncertainties in numeric terms, and discuss these here. The words "substantially" and "higher" seem too abstract.*
  We find it difficult to provide a general numeric uncertainty here, as individual freezing curves are substantially different from one another. Moreover, we cannot concisely predict how the cumulative INP concentration of a freezing curve would extrapolate at lower temperatures. Further, as stated in the text the underestimation is dependent on the temperature were the last droplet froze. For example, if sample a. was completely frozen at -24°C and sample b. at -28°C, the extrapolated cumulative INP concentration at -30°C would likely be much higher at sample a.

- *P8L2: The authors may want to recap the unique importance of 1977, 1680 & 1630 and provide the readers a brief justification of why they were picked for SEM analysis.*
  Unfortunately, as this was a novel measurement approach for us, the labor intensive SEM analysis was limited to a small number of samples in this study. We plan to increase the number of SEM samples future studies.

The three samples were selected for different reasons. The 1977 sample was selected as example for an IN active modern-era sample. The 1680 sample was selected, because it had an average INP concentration at -25°C and was in the middle of the time series. The 1630 sample was selected, because it showed extraordinarily high INP activity and we were interested to find out if we could identify the underlying reasons in the chemical aerosol signature of the sample. We will add a short paragraph to the manuscript:

"The 1977 sample was selected exemplarily as an active modern-era sample. The 1680 sample was chosen for its average INP concentration at -25 °C, as well as being in the middle of the time series. The 1630 sample was analyzed with SEM, because it had an extraordinarily high INP activity at comparably warm temperatures."

- *Sect. 2.7: Briefly describe the operation conditions of SEM-EDX – beam intensity, WD, SS etc. Were these experimental variables all consistent for all analyses?*

    Yes, experimental variables were consistent for all three samples. We expanded the method description, which now reads:

    "For SEM-EDX analysis on each filter some 100 rectangular fields of about 100 µm x 100µm in the center of the filter were scanned and for all detected particles the size was determined and an EDX analysis (acceleration voltage: 20 kV, spot size: 4, acquisition time: 10 s, working distance: 10 mm) was performed. Using this procedure, particles down to approximately 250 nm were detected. Smaller particles will often be overlooked. This is also true for larger carbonaceous particles, because of their poor contrast on the polycarbonate filter."

- *P8L11-12: The reviewer suggests excluding "Smaller particles will…". Adding not much value to the section.*

    As reviewer 3 seemed to be confused which particles were collected for SEM analysis and which were likely lost during filtering, we think we should keep this sentence.

- *P8L16: review the state of the art of → reviewed several*

    Okay.

- *Sect. 2.8: In general, this section can be much more concise. Especially, P8L30-P9L2 seems containing repetitive information and, thus, could be excluded. Three most important sentences in this section are: P9L6 However,…; P9L7 Unfortunately,…; and P9L15 Therefore,… The reviewer suggests the authors to summarize the section by putting simple emphasis on these, and reduce the # of words. The reviewer defers to the point addressed in P9L18-21. No worries. The authors' method sounds.*

    We agree with the reviewer that the highlighted text passages are the most important sentences in this section. We understand that the presented approach of listing and addressing potential factors influencing the background signal is not strictly necessary and it may take the reader some time to find the most relevant bits concerning the actual measurement data. But we feel in that these uncertainties and measurement routines are often not clearly addressed in many publications. Also, this is the first first-author publication concerning the FRIDGE droplet freezing method, and we would like to be able to refer to this

method paper in our future publications. Therefore, we chose to be rather detail-oriented.

- *P10L19-20: This means as well...* → *This implies that INP concentrations may be higher in ice core samples than ambient INP concentrations at any given time. Or something similar?*

  We rephrase the sentence to:

  "This implies that INPs may be overrepresented in ice core samples compared to non-INPs or the ambient atmosphere at any given time."

- *P10L26-27: The reviewers agrees about INPs being preserved. The authors may add discussion of Beall et al. (Beall, C. M., Lucero, D., Hill, T. C., DeMott, P. J., Stokes, M. D., and Prather, K. A.: Best practices for precipitation sample storage for offline studies of ice nucleation, Atmos. Meas. Tech. Discuss., https://doi.org/10.5194/amt-2020-183, in review, 2020.).*

  We thank to reviewer for the suggested literature, which we were not aware of. Thanks to this note and a concern shared by reviewer 2 and 3, we now feel that we need to highlight possible losses of (high temperature) INP activity due to storage effects. Although we ensured a frozen storage at our laboratory, the samples needed to be melted and refrozen several times prior to the IN measurements, possibly deactivating warm INPs and thus lowering the cumulative INP concentration. Text passages were added to:

  Page 9, line 30: "However, recent studies indicate that sample storage (i.e. storage temperature) significantly affects the ice nucleation activity of fresh precipitation samples in the range of -7 C to -19 °C (Beall et al., 2020). For example, samples stored at room temperature lost on average 72% of their INPs compared to the freshly analyzed samples. An average INP loss of 25% was still observed, even when samples were stored at -20 °C. Storage time did only weakly affect the INP concentrations. Therefore, based on this study a loss of INP activity on the order of a factor of 2 – 5 is possible, if not likely for the ice core measurements presented here. Furthermore, it is likely that the warmer end of INPs were disproportionally affected by these disturbances, while cold-temperature INPs were likely more robust. However, as all the samples experienced the same sample history, relative changes within the ice core can still be interpreted."

  Page 10, line 27: "However, as previously stated, storage conditions may have affected the INP activation."

- *P10L31-34: Not adding much value to the section. The reviewer suggests removing this part from the manuscript.*

  We recognize that the quote does not add much to the manuscript. Although, we like the quote, we deleted the sentence from the manuscript.

- *P11L3: Where does this 'an order-mag.' come from? Please clarify in the text for the readers.*

  We think we give ample explanations in sections 2.9 and 2.10 why the conversion to atmospheric concentrations is uncertain. The phrase signaling that the conversion "should be interpreted only as an order-of-magnitude estimation", is added to the text to highlight these uncertainties and to sensitize other researchers, which might want to use the data for atmospheric modelling, etc.

- *P13L3: The reviewer accepts the idea of conversion. If the authors are confident it is only +/- 50%, the reviewer suggests massively cut # of texts/words in this section. In general, this section is hard to follow. Spending full 2 pages to derive seems a simple sub-conclusion (i.e., P13L6-7) seems overwhelming. You may list the typical value of each variable (A, v_dry, and epsilon) +/- 'reasonable' upper/lower ranges (that correspond to shape a blue square in Fig. 4) in a table format to reduce # of total words. For that matter, the reviewer wonders if Fig. 4 is really needed and meaningful. A different presentation (again, tabular format) may be considered.*

  We shortened the text substantially and moved the extended version into the Supplement. However, we feel that a visual representation of the conversion factor depending on dry and wet deposition gives the reader a good idea about the sensitivity of those uncertain parameters. Therefore, we prefer Fig. 4 to a table and chose to keep it.

- *P13L13-15: This paragraph seems not fitting here.*

  Okay.

- *P13L28: very steep freezing → local maximum in – or something similar*

  We chose to keep the phrasing.

- *P13L29-30: → We verified a reproducibility of our results by confirming two separate measurements agreed each other. This verification eliminated the contamination during our FRIDGE measurements. Does this what the authors mean?*

  We rephrased the sentence. The manuscript now reads:

  "We verified our results by reproducing the measurement of this sample. The separate second measurement confirmed the strikingly different freezing characteristics, thus eliminating a contamination during the FRIDGE measurements themselves."

- *P14L4: → they showed a frozen fraction of only 0.7% on average.*

  Okay.

- *P14L7: average in ice concentration → average $N_{INPice}$*

  Okay.

- *P14L9: Here → At this temperature,*

  Okay.

- *P14L10: From here onward,... ➔ Next, our characterization of INPs at -25°C is specifically discussed.*
  We rephrased the sentence. The manuscript now reads:
  "Henceforth, the discussion of results is focused on the characterization of INPs at -25°C specifically."

- *P14L11-12: ...every single sample... ➔ all samples showed some droplet freezing events at this T.*
  Okay.

- *P14L14: , so the reader can see ➔ in order to clarify*
  We rephrased the sentence. The manuscript now reads:
  "[...] in order to illustrate the typical variation in the INP concentration, while still allowing for easy identification of differences in the absolute INP concentration level."

- *P14L15: The reviewer suggests deleting ", but is still..." – not much value added.*
  See above.

- *P14L16: arise from ➔ can be inferred from*
  Okay.

- *P14L17: delete "somewhat" and specify/clarify what include "more recent samples" in the main text.*
  Okay. The manuscript now reads:
  "We find on average higher and more variable INP concentrations for the last couple of decades as compared to the rest of the time series."

- *P14L22: Yet, ➔ Nevertheless,*
  Okay.

- *P14L30 moderate yet significant ➔ notable*
  We kept the phrasing as is, because we think our phrasing is more precise.

- *P14L34-P15L2: Delete 'however' and re-write the sentence to clarify what the authors mean to the readers.*
  We rephrased the sentence and moved Tab. 2 to the Supplement:
  "When the data is grouped into subsets according to Fig. 1, we find that the correlation weakens for the 10 year samples and the modern day samples, but increases for special event and seasonal samples (Tab. S2)."

- *P15L6: We like to point out here ➔ It is noteworthy*
  Okay.

- *P15L11: That being said, going forward ➔ Regardless,*
  Okay.

- *P15L12-14: "The four…" – the reviewer could not understand what it meant. Please rephrase and clarify the sentence.*

  The sentence was added to clarify that samples with the years 1960, 1970, 1980 and 1990 originally belonged to both subgroups (10 years and 1960 to 1990), but in the analysis presented they are only included in the 1960 to 1990 subgroup. We now see that the sentence is redundant and confuses the reader more than it helps, so we removed the sentence from the manuscript.

- *P15L14—16: →  The observed difference between pre- and post-1960 samples is based on Subramanian (2019), which defines the 20th century as the beginning of the Anthropocene. Keep it simple, and delete "Note, however,…" – not much adding in.*

  The above cited literature is an article from the News Feature from Nature and does only report about the scientific debate about when to start the Anthropocene. We now added a solid reference. The manuscript now reads:
  "The observed difference between pre- and post-1960 samples is based on Zalasiewicz et al. (2011), who propose to define the middle of the 20th century as the beginning of the Anthropocene."

- *P15L17-19: But, then, excluding it also biases the authors' data… It is an important outlier, correct? It can be still excluded, but the reviewer suggests the authors to provide a better (and more constructive) justification to exclude it in the text.*

  We changed the text to:
  "Furthermore, we excluded the sample from 1630 in most of the following analysis in favor of more consistent freezing spectra. The statistical outlier is certainly important, as it was the only sample that was completely frozen before reaching -22 °C. At this state, however, we cannot explain what caused its high IN activity (cf. section 3.2). Moreover, as stated previously a contamination prior to the INP analysis cannot be excluded completely for this sample. Including the outlier does not change the general results."

- *P16L1: delete "seem to"*
  Okay.

- *P16L7: Only 36 particles for 1977. Please provide a justification for this small #.*
  On this filter only a very small number of particles were detected during the SEM analysis in the analyzed filter center region. The border regions were not analyzed, because of a higher risk of artefacts. The reason for the very low number is not known.
  Page 16, line 15 now reads:
  "For example, only a very small number of particles were detected on the 1977 sample in the analyzed center region of the filter. Generally, the border regions were not analyzed due to a higher risk of artefacts."

- *P16L11-13: Please provide reference(s). "will be feldspars" sound awkward. Please rephrase it.*
  Although EDX analysis does not allow an unambiguously mineralogical phase assignment, the typical elemental ratios (e.g. Al/Si ratio) and the content of minor elements in the alumosilicate particles allow at least an appraisal of the present silicate classes. In this case most of the detected alumosilicates are most

likely feldspars (sodium and potassium feldspars), amphiboles and pyroxenes. Besides this, some quartz and clay minerals are also present. For a more profound and detailed phase classification Transmission electron microscopic (TEM) investigations could be performed. Such kind of investigations require a specific sample preparation and are very time intensive.

We rephrased the mentioned sentence to:

"[...] most of the detected alumosilicates are most likely feldspars (and here more sodium and potassium feldspars), amphiboles and pyroxenes."

- *P16L16: How did the authors define "fly ash" through SEM-EDX? Reference(s)?*

  The fly ash definition in SEM analysis is strictly following the morphological analysis. If "perfect" melting spheres of refractive particles are detected the particles will be classified as fly ashes. For particle types with high melting points (silicates, metal oxides) no other particles source (as the high temperature process producing fly ashes) is known producing "perfect" spherical particles beside volcanic activities. But even when particles from volcanic activity can also show "spherical-like" morphologies, they differ strongly in morphology and mixing-state from fly ashes.

  However, not all fly ashes are spherical, therefore not all fly ashes, but only the spherical ones, can be classified/identified in SEM analysis.

- *P16L17: No notable difference found here might be due to limited # of particles analyzed, correct? If so, it should be stated in the text.*

  Correct. The manuscript now reads:

  "Otherwise there was no obvious distinction between the modern-era sample and the other two samples with regards to their chemical composition, which might be due to the limited number of particles analyzed."

- *P16L27-28: does seem to follow → shows*

  Okay.

- *P16L33-P17L7: The reviewer suggests the authors to soften the tone regarding the annual cycle. Yes. It is nice to see the seasonal cycle exists in this subset of samples, but the authors might need to be careful on not generalizing it as a bold conclusion here. The authors need to make it clear in the text in this particular section that this applies to only what they have analyzed for. Otherwise, please provide a proper justification why the authors believe the seasonal cycle could persist for other eras.*

  Okay. The manuscript now reads:

  "These findings suggest that the INP concentration in this year was subject to the annual dust input in Greenland. As the seasonal variability in particulate dust number can be clearly detected throughout the entire core, we expect that such a seasonal INP variability will hold for the entire record. Future high-resolution studies will have to test this assumption. Bory et al. (2002) show that the main dust source in Northern Greenland is the Taklamakan Desert in Northern China. At the beginning of the monsoon season, the dust particles are transported to Greenland within a few days via the jet stream and cause the annual maximum dust input for Greenland in spring."

- *P17L1: How about an episode of dust along with Atlantic Monsoon? How about Iceland etc.? Suggested reading: Iceland is an episodic source of atmospheric ice-nucleating particles relevant for mixed-phase clouds, Sanchez-Marroquin, https://advances.sciencemag.org/content/6/26/eaba8137.abstract*

  We thank the reviewer for suggesting the interesting article. We agree that both Saharan and Icelandic dust are possible (episodic) contributors to atmospheric INPs reaching the Arctic. However, the literature consensus suggests that the listed East Asian deserts and the described mechanism are largely responsible for the dust input in Greenland. Nevertheless, we added a sentence to the manuscript:

  "Furthermore, episodic dust transport from the Sahara desert (Lupker et al., 2010) and Iceland (Sanchez-Marroquin et al., 2020) may have contributed as well."

- *P17L30-P18L4: The reviewer appreciates the authors being careful, honest scientists by these statements here and elsewhere in the manuscript. Nonetheless, this part (right before the conclusion!) may give a very negative impression about the authors' study to the readers. Scattered concern statements throughout the manuscript bothers this reviewer, at the least. The authors may compile their concerns here and there regarding all uncertainties in Sect. 2.9 in a brief manner. The readers would understand that the results come with uncertainties, and the authors do not need to be too sensitive to sound.*

  We did not want to overly interpret the data as there are a number of uncertainties. Therefore, we were cautious when stating and discussing the findings in the manuscript. However, we understand the point the reviewer is trying to make.

  Regarding the issue of a potential post-coring contamination, we now reanalyzed some existing Abakus (particle diameter >1 µm) and SPES (single particle extinction and scattering instrument, particle diameter <1 µm) ice core data, which we like to share. The Abakus was used on the B17 and the EGRIP S6 ice core. On the S6 core we also used the SPES instrument, which was not yet available, when B17 was measured:

  Abakus: Analyzing the Abakus data of those two independent ice cores we find an average twofold increase of the mean background concentration of particles larger than 1 µm (but also in $Ca^{2+}$) in the top 8 m, which roughly corresponds to the time interval of 1960 – 1990, compared to deeper / older data. Already at 20 – 30 m (around 1900), where the firn is still porous, we do not see such an increase. The seasonality in the top 8 m is not as clean as in deeper intervals, which is to be expected as there are lots of breaks and wicking effects within the top 8 m. However, the seasonality is still detectable and comparable in amplitude to intervals below 20 m or even below 60 m (solid ice). Obviously, we cannot rule out contamination effects with absolute certainty, but the existence of a distinct seasonal variation is a valid argument that the observed increase in the dust concentration may be atmospheric.
  We will try to resolve the seasonality of the porous firn in future INP studies.

SPES: The SPES data from the S6 ice core looks quite different. For particles smaller than 1 µm we observed the average background concentration to increase by a factor of 4 − 5 within the top meters of the ice core. Furthermore, there was no seasonal signal within the top 100 years of the S6 ice core. Therefore, we conclude that for these smaller particles the post-coring contamination of the porous firn is severe. At this state, we do not know what kind of particles they are, but the mean diameter of the number size distribution is about 0.6 µm.

In conclusion, the INP results seem to agree sufficiently to the observations made by the Abakus, which sees an average twofold increase for the 1960 − 1990 interval in B17 (and S6). One could cautiously argue that therefore the INPs seem to reflect the mineral dust input of particles larger than 1 µm. However, as we have seen by the SPES data (from another ice core), a contamination effect is likely for particles smaller than 1 µm (and cannot be excluded completely for larger particles). As we did not observe the 4 − 5 times increase in INP concentrations as the SPES did for particles smaller than 1 µm, we expect that these contamination particles are no particularly active INPs in the investigated temperature regime, either due to their size, which might be substantially lower than 1 µm, or their surface structure, morphology or chemical composition, etc.

Regardless, we carefully read the manuscript again and removed some repetitious sections that mentioned a possible post-coring contamination, while adding to other text passages. Below we now list each instance, where the possible effect was mentioned, and describe if the passage was kept, removed or changed.

Abstract: "[...] or some post-coring contamination of the topmost, very porous firn."
We removed the text passage from this section.

Section 2.9: The effect was not yet mentioned.
We now introduce the effect in Section 2.9 (Other uncertainties).
Page 10, Line 4 now reads: "Specifically, we like to emphasize that the topmost part of the ice core is made up of relatively porous firn, which is more prone to post-coring contamination of dust during storage as compared to the rest of the ice core. Preliminary results of two particle counters (Abakus: spherical diameter > 1 µm, SPES: spherical diameter < 1 µm) from the B17 ice core (only Abakus) and the EGRIP S6 ice core (Abakus and SPES, 75.62° N, 35.97° W, 2702 m asl, C. Zeppenfeld, personal communication) suggest that a contamination effect is likely for particles < 1 µm and rather unlikely for particles > 1 µm. However, post-coring contamination still cannot be fully excluded for the latter measurements."

The effect was introduced in Section: 3.1: "It is noteworthy that the topmost part of the ice core is made up of relatively porous firn, which is more prone to post-coring contamination of dust as compared to the rest of the ice core. Unfortunately, we cannot entirely exclude the possibility that differences emerged or are enhanced due to post-coring contamination of the firn, as the ice core was stored for some time, despite the CFA decontamination technique."

We now only mention the effect shortly here.

Page 15, Line 7 now reads: "Unfortunately, despite the CFA decontamination technique we cannot entirely exclude the possibility that differences emerged or are enhanced due to post-coring contamination of the porous firn, as the ice core was stored for some time. Preliminary measurements (cf. section 2.9) found a twofold increase of particles larger 1 µm in the top 8 m (roughly the time interval of 1960 – 1990) compared to older intervals, which does seem to match the results observed by the INP measurements. Further, a distinct seasonality could be established for the dust measurements of the top layers, which argues against a strong contamination effect."

Later in 3.1 (after the revision): "If there was, however, only little influence by post-coring contamination and the latter two listed effects, the findings suggest that certain particles that are ice nucleation active in a mid-supercooled temperature regime may be more abundant in today's atmosphere."

We kept the mention of the effect here to transition to the next topic (i.e. which INPs could be enhanced in today's atmosphere).

Section 3.4: "However, we cannot fully rule out post-coring contamination as the cause for the observed differences."

We removed the text passage from this section.

Conclusions: "Alternatively, differences may have been caused by post-coring contamination, which is likely more relevant for these samples as they stem from the more porous firn layer."

We removed the text passage from this section.

- *P17L23: Fig. 11 tells the reviewer that the diversity may derive from the concentration and size of dry & wet deposited particles rather than the listed differences? The variability due to composition is ruled out in Sect. 3.2, correct? Please clarify.*

    The stated line gives possible explanations why our range and means of absolute INP concentrations are different from what Hartmann et al. (2019) observed in two Arctic ice cores. After reading the comments of all reviewers, we added storage effects to the list. Then, as we understand the reviewer refers in the comment to the diversity of INP concentrations that is observed in our data set specifically, which is another matter. We agree that the concentration and size of deposited particles are likely a driver of the INP concentration in the ice core. However, we don't think that the results of section 3.2 definitely rule the chemical composition of particles out here. We explicitly say that "[...] due to the low number of analyzed particles, we were unable to determine significant differences in particulate composition of the particles and size distribution in the three samples." The sentence now reads:

    "This disparity may arise from experimental (droplet volume, etc.), methodological (e.g. sample storage conditions) and or geographical differences, which may affect the deposition mechanisms and efficiency"

- *P18L11: particularly → significantly or substantially?*

    We chose the keep the phrasing.

- *P18L12: group → selected subset*
  Okay.

- *P18L14: recap and specify "certain aerosol species" here for the readers.*
  Okay. The manuscript now reads:
  "Furthermore, we found significant correlations between concentrations of INPs and the insoluble particle concentration > 1.2 µm, Ca2+ concentration and the conductivity for a broad range of temperatures."

- *P18L20: Delete "several mechanisms can be considered by which". The sentence makes sense without it.*
  Okay.

- *P18L31: The reviewer strongly agrees ☺*

- *P19: Perhaps, one of top priorities for the future ice core INP research includes the assessment of particle size distributions in liquid samples by DLS etc. The authors may elaborate it as an outlook? Connecting INP properties to aerosol propensities may resolve some raised concerns?*
  We agree with the reviewer that the proposed method of dynamic light scattering to gain information about the particle size distribution in liquid aerosol samples is certainly very interesting and promising. However, as far as we know, this method has not yet been tested in ice core studies. Furthermore, the low number of particles might be challenging for the instrument. We added a sentence to the manuscript:
  "[...] Furthermore, we plan to expand our ice core analysis to include a more rigorous, systematic study analyzing the chemical and morphological composition of insoluble aerosol particles as well as their size distribution by scanning electron microscopy. Particle size distributions of liquid samples may also be attainable by the dynamic light scattering and the single particle scattering and extinction method."

- *Tables 1 & 2: Add "Temperature (°C)" as the first column header, and delete °C from the send row.*
  Okay.

- *Table 2: What are "dust, volcanic and seasonal"? Please clarify within the table caption.*
  We added a reference to Fig. 1 (and moved Tab. 2 to the Supplement).

- *Fig. 1 caption: → Time coverage of the samples selected for assessing IN properties.*
  Okay.

- *Fig. 1 caption: longer → long*
  Okay.

- *Fig. 2: Adding the least active spectrum from the core sample (P9L14) may increase the visual importance of this figure.*
  Okay.

- *Fig. 3: The authors may superpose the 1:1 ratio line on top of the fit line. Doing such reinforce the authors' point in a visible manner.*
  Okay.

- *Fig. 6: INP [L^-1_ atm] or N_INP_atm? Perhaps, the authors may choose one way to improve the consistency throughout the manuscript.*
  Okay.

- *Fig. 7 caption: "However, data..." – the reviewer did not understand this. Please clarify.*
  The sentence points out that the underlying data (i.e. 31 samples that averaged over time periods of 0.24 years to 1.96 years, average 0.58 years, see Fig. 1b) was interpolated in time to create visually regular yearly columns of frozen fraction data, similar to what is presented in Fig. 5. After the interpolation 30 yearly columns are shown with the same width. We added the original (non-interpolated) data here and in the supplement (Fig S.2) for clarification. The caption of Figure 7 now reads:
  "However, data points are interpolated in time to generate yearly columns of regular width. The non-interpolated data is presented in Fig. S2."

[Figure]

- *Fig. 7 caption: Delete "Note, that".*
  Okay.

- *The reviewer enjoyed reading it. Hope some of suggestions/comments made here help the authors.*
  We are glad that the reviewer liked the manuscript. Again, we thank the reviewer for their valuable suggestions, which will definitely improve the paper.

Literature

Beall, C. M., Lucero, D., Hill, T. C., DeMott, P. J., Stokes, M. D., and Prather, K. A.: Best practices for precipitation sample storage for offline studies of ice nucleation, Atmos. Meas. Tech. Discuss., https://doi.org/10.5194/amt-2020-183, in review, 2020.

Bory, A. J.-M., Biscaye, P. E, Svensson, A., and Grousset, F. E: (2002): Seasonal variability in the origin of recent atmospheric mineral dust at NorthGRIP, Greenland, Earth and Planetary Science Letters, 196, 3–4, 123–134, https://doi.org/10.1016/S0012-821X(01)00609-4, 2002.

Chen, J., Wu, Z., Augustin-Bauditz, S., Grawe, S., Hartmann, M., Pei, X., Liu, Z., Ji, D., and Wex, H.: Ice-nucleating particle concentrations unaffected by urban air pollution in Beijing, China, Atmos. Chem. Phys., 18, 3523–3539, https://doi.org/10.5194/acp-18-3523-2018, 2018.

Hartmann, M., Blunier, T., Brügger, S. O., Schmale, J., Schwikowski, M., Vogel, A.,Wex, H., Stratmann, F.: Variation of ice nucleating particles in the European Arctic over the last centuries, Geophysical Research Letters, 46, 4007–4016, https://doi.org/10.1029/2019GL082311, 2019.

Lupker, M. Aciego, S. M., Bourdon, B., Schwander, J., and Stocker, T. F.: Isotopic tracing (Sr, Nd, U and Hf) of continental and marine aerosols in an 18th century section of the Dye-3 ice core (Greenland), Earth and Planetary Science Letters, 295, 1–2, 277–286, https://doi.org/10.1016/j.epsl.2010.04.010, 2010.

Murray, B. J., O'Sullivan, D., Atkinson, J. D., and Webb, M. E.: Ice nucleation by particles immersed in supercooled cloud droplets, Chem. Soc. Rev., 41, 19, 6519–6554, https://doi.org/10.1039/C2CS35200A, 2012.

Sanchez-Marroquin, A., Arnalds, O., Baustian-Dorsi, K. J., Browse, J., Dagsson-Waldhauserova, P., Harrison, A. D., Maters, E. C., Pringle, K. J., Vergara-Temprado, J., Burke, I. T., McQuaid, J. B., Carslaw, K. S., and Murray, B. J.: Iceland is an episodic source of atmospheric ice-nucleating particles relevant for mixed-phase clouds, Science Advances, 6, 26, https://doi.org/10.1126/sciadv.aba8137 , 2020.

Subramanian, Meera: Anthropocene now: influential panel votes to recognize Earth's new epoch, https://www.nature.com/articles/d41586-019-01641-5, last access: 20 February 2020, 2019.

Zalasiewicz, J., Williams, M., Haywood, A., and Ellis M.: The Anthropocene: a new epoch of geological time?, Phil. Trans. R. Soc. A., 369, 835–841, https://doi.org/10.1098/rsta.2010.0339, 2011.

Zhao, B., Wang, Y., Gu, Y., Liou, K.-N., Jiang, J. H., Fan, J., Liu, X., Huang, L., and Yung, Y. L.: Ice nucleation by aerosols from anthropogenic pollution, Nat. Geosci., 12, 602–607, https://doi.org/10.1038/s41561-019-0389-4, 2019.

---

## Author Comment (AC2) · 9 Sep 2020

*The authors have made a great effort in trying to reconstruct from the analysis of an ice core the atmospheric concentration of ice nucleating particles (INPs) in the atmosphere over Central Greenland between the years 1370 and 1990. It is only the second such attempt, after a similar but less comprehensive study published last year by another group. Overall, the manuscript is clearly written. Everything is well explained. The text is easy to follow. Data are arranged in a meaningful way in Tables and Figures. Half of the main text is description of methods. Interpretation of results is cautious, if not hesitant. It is here that I see some room for improvement, apart from a few other, minor issues.*

*The most surprising outcome of this study, from my point of view, is the narrow range of INP concentrations in ice and atmosphere during a period in which Earth has seen a tenfold increase in land area used for agriculture (Pongratz et al., 2008). Ploughing of the North American prairies and the Russian steppes during the past two centuries has greatly accelerated wind erosion with drastic consequences, like the harvest failure of 1891 in the Russian steppes (Moon, 2005) and the Dust Bowl situation in the USA during the 1930s. Also intensive grazing by exploding numbers of domesticated animals has had its share in fostering wind erosion during that time (Neff et al., 2008). Other than desert dust, soil dust from more fertile land carries INPs active at moderate supercooling (O'Sullivan et al., 2014). Therefore, I would have expected to see growing number concentrations of INP active at temperatures at around -15 °C or warmer in samples deposited over the last two centuries. However, this does not seem to be the case. Only 3% of all samples, each consisting of 0.5 mL of melted ice, contained INPs active at -15 °C. There are at least two plausible explanations for this observation. First, it could be that anthropogenically caused dust in the midlatitudes was not transported in detectable quantities to the Arctic and deposited in Central Greenland. The overwhelming majority of dust and INPs deposited in the Arctic probably originates from regions north of 60 °N in America and Eurasia, latitudes not much affected by landuse change in the past. Regions in North America located south of 60 °N probably contribute less than one percent to the total surface dust concentration in the Arctic (Groot Zwaaftink et al., 2016, Table 3 therein). Thus,*

large-scale landuse change and increased wind erosion of fertile soils in the midlatitudes following the colonisation of North America by settlers mainly from Europe may indeed not have had a marked effect on INPs deposited in Central Greenland, although it clearly increased dust deposition in the midlatitudes (Neff et al., 2008).

> We greatly appreciate the interesting and well-presented, detailed remarks, and suggested literature by the reviewer regarding the potential anthropogenically enhanced source of soil dust INPs. We fully agree to the statements made by the referee. We also list soil and desert dust due to desertification, land-use change, the expansion of agriculture and related practices, and consequently higher erosion rates as one of our main candidates for anthropogenic INPs (Page 3, lines 6 – 24). We added a paragraph to the discussion in section 3.1 (see following answer).

Another explanation for landuse change over the past centuries not being reflected in the INP record of the analysed ice core could be a loss of IN-activity, in particular the loss of biological INPs that dominate the spectrum at temperatures warmer -15 °C in ice or during sample preparation, in particular during melting of the core and while the samples were in liquid form. Since Hartmann et al. (2019) found clearly enhanced INP activity between -5 °C and -15 °C even in some older sample (year 1484), sample preparation may be the more relevant issue. It would be interesting to know the temperature on the hot side of the instrument in which the ice core was melted. Further, for how long, in total, were samples in liquid form between the first melting of the core and INP analysis? Evidence pointing at a partial loss of INPs is in Figure 9 of the manuscript in discussion. It shows from -30 °C to -24 °C increasingly larger INP concentrations in the modern, as compared to the older samples. The relative difference between modern and older samples collapses quickly towards the warmer end of the temperatures scale. I would have expected this difference to continue increasing further until the warmest temperature is reached at which INPs are detectable. Maybe there is no difference at warmer temperatures detectable today because INPs active above -22 °C had lost their activity before INP analysis? In my experience, any challenge put to a population of INPs, such as warming or storage in water, always leads first and foremost to a loss of those INPs that are active at the warmest temperature. The warm temperature "bulge" in a cumulative INP spectrum disappears with increasing severity or duration of a challenge, resulting in the cumulative spectrum approaching a linear shape on a log-scale. The same applies to certain mineral INPs (Harrison et al., 2016, their Figure 4a, top panel). Partial deactivation most likely results in the remaining part of the INP population becoming increasingly homogenous, a guess supported by Figure 8 in the discussed manuscript: the distribution of frozen fractions at a specific temperature was much narrower for the older samples (pre- 1960) as compared to the modern samples (1960 to 1990). The majority of fragile INPs, which may have been present at the time of deposition, and still are to some extent in the samples from 1960 onwards, may have been lost, leaving behind a relatively homogenous population of very stable INPs. To summarise, very limited dust transport from the midlatitudes, where most landuse change has happened in past centuries, and deactivation of INPs active at temperatures warmer than -20 °C may explain why the concentration of INPs (Murray et al., 2012). Deactivation might have happened during decades and centuries in the ice core is confined to a narrow range and does not reflect the growing human impact on land over the past few centuries. These considerations are of cause speculative, but I hope they encourage the authors to push their interpretation a bit further.

> We thank the referee for their interesting and inspiring thoughts, and the encouragement to deepen our interpretation of the data. Both ideas presented by the reviewer appear probable to us. The reviewer's argument for a partial deactivation of

the warmer end of INPs from the interpretation of the freezing spectra and INP concentrations in Figs. 8 and 9 are convincingly portrayed. As a consequence we retraced the conditions of storage and melting of the samples as well as possible. In fact, we come to the conclusion that the suggested partial deactivation of INPs is at least possible. Although care was taken during and before INP measurements were made in our laboratory, sample vials may have been subject to temperatures between 0°C and room temperature for up to some tens of hours in total (during repeated cycles of melting and refreezing, supporting measurement, and transport). For example, during the CFA decontamination step the ice core is melted in the typically temperature range from 12 – 25 °C depending on the density of the ice and the desired meltspeed. Unfortunately, the exact times and temperature conditions during storage, etc. are difficult to assess in retrospect. We recognize the need to decrease the time samples were in an unfrozen state in future studies.

We added the following text passages to the manuscript:

Page 5, line 11: "Hence, it was ensured that samples remained frozen at all times in our laboratory. However, sample vials may have been subject to temperatures between 0 °C and room temperature for up to some tens of hours in total (during repeated cycles of melting, storage and refreezing, non-INP measurements, and transport, etc.)."

Page 9, line 30: "However, recent studies indicate that sample storage (i.e. storage temperature) significantly affects the ice nucleation activity of fresh precipitation samples in the range of -7 °C to -19 °C (Beall et al., 2020). For example, samples stored at room temperature lost on average 72% of their INPs compared to the freshly analyzed samples. An average INP loss of 25% was still observed, even when samples were stored at -20 °C. Storage time did only weakly affect the INP concentrations. Therefore, based on this study a loss of INP activity on the order of a factor of 2 – 5 is possible, if not likely for the ice core measurements presented here. Furthermore, it is likely that the warmer end of INPs were disproportionally affected by these disturbances, while cold-temperature INPs were likely more robust. However, as all the samples experienced the same sample history, relative changes within the ice core can still be interpreted."

Page 10, line 26: "Hartmann et al. (2019) come to the same conclusion that INPs are well preserved in an ice core and a reconstruction of their concentration for past climates is possible. However, as previously stated, storage conditions may have affected the INP activation."

Page 16, line 5: "Considering that the total global agricultural land area is estimated to have increased by a factor of 10 from 1400 to 1992 (Pongratz et al., 2008) combined with the fact that wind erosion has immensely accelerated within the last two centuries (Neff et al., 2008), partly due to intensive grazing by the heavily increasing number of domesticated animals, one could even have expected larger differences between the two data groups. Especially in the temperature range around -15 °C, at which soil dust INPs from fertile agricultural regions are known to be active (O'Sullivan et al., 2015). We can only speculate why we generally did not observe many INPs in this temperature range, and why the significant differences between the two data groups were only observed for temperatures below -22 °C. First, it is possible that dust from anthropogenic practices was not transported to Central Greenland in a detectable

amount. According to Groot Zwaaftink et al. (2016), most of the dust input contributing to the dust surface concentration of the Arctic is from Eurasia north of 60° N, North America north of 60° N and Asia south of 60° N. In contrast, North America and Europe south of 60° N, where land-use change and the agricultural expansion are most prominent, contribute only little to the Arctic dust input (below 1%). Moreover, Asian agricultural dust sources may not exhibit the necessary high wind speeds to inject mineral dust into the upper troposphere as required for long-range transport to Greenland. In contrast, mineral dust from the Taklamakan desert is intrinsically linked to dust storms in this area.

Second, the more fragile (biological) INPs may have been deteriorated during sample storage (Beall et al., 2020). As a result, the warm-end of INPs might have been largely lost, leaving only a homogeneous fraction of very stable INPs behind. Figures 8 and 9 present some evidence for this hypothesis. As seen in Fig. 8, we find a much narrower range of frozen fractions for the 10 year samples, hinting at a rather homogenous population of INPs. On the other hand, the variability is much higher for the modern-day samples, possibly because some of the more fragile INPs were still active. However, as both sample groups experienced the same sample history after coring, this hypothesis would only be reflected by deterioration effects related to the time elapsed since the particles were deposited in the ice. Furthermore, Fig. 9 depicts increasingly greater relative differences in the INP concentration from -30 °C to -24 °C until the warmer end of the data is reached, at which only few samples show ice nucleation activity. This observation could possibly be explained by assuming that the warmer INPs were largely deactivated due to storage effects."

*Minor issues*

- *Page 9, lines 19-20: I am always at a loss when told that results "...should be interpreted with care." Is not every interpretation or conclusion based on empirical evidence a preliminary one and absolutely true statements only to be found within closed systems (mathematics, logic)?*

   The reviewer is obviously right with their remark. The phrase in question is only added to sensitize the reader about the lower temperature part of the data, because we did not subtract a background as is otherwise common practice (Page 9, lines 6 – 17).

- *Page 9, line 32: Why use the number of frozen droplets and not the number of INPs in the assay (INPs in 195 droplets) as the criterion from which to estimate uncertainty?*

   We believe both phrasings mean the same thing. The most active INP within a droplet will initiate the freezing of said droplet. Therefore, the number of frozen droplets is related to the number of active INPs. The uncertainty specifies an upper and lower range in the number of droplets that are expected to freeze within a 95% confidence interval at a certain temperature.

- *Page 15, line 26: The data has a lognormal distribution. Was it log-transformed before the t-test?*

   The data was not log-transformed in the reviewed version of the manuscript. Transforming data yields similar results: We observed significant differences in

the average (log-transformed) INP concentration for -23 °C (p < 0.00011), -24 °C (p < 0.000002), -25 °C (p < 0.00005), -26 °C (p < 0.0011) and -27 °C (p < 0.02).

- *Conclusions section: Effects of INPs on cloud radiative properties are mentioned and I wonder whether the very small number concentrations found in the ice core, and the difference between 1960 to 1990 or before, are indeed in a range where they might lead to differences in radiative properties?*

  Answering this interesting question is beyond the scope of this paper. However, we believe it is worth exploring in future studies, e.g. by atmospheric modelling (page 19, line 13 – 15).

- *Regional sources and geographical differences in INPs may not only be accessible through the analyses of ice cores but also through modelling approaches making use of historical records of land cover.*

  We agree with the reviewer and refer to the very last sentence of our manuscript (Page 19, lines 13 – 15). The addition of modelling studies based on historical records of land cover definitely seems like an interesting approach, which we did not consider before. We modified the manuscript, which now reads:
  "Finally, a modeling study could help identify (possibly anthropogenically altered) INP source regions (e.g. based on historical records of land use cover) and estimate the potential atmospheric impact that could be expected from a threefold increase of INPs at −24 °C since the mid of the twentieth century, as it was seen in this study."

- *Figure 1b: Would it be possible to indicate the season for samples with time coverage below one year?*

  Unfortunately, the dating of the ice core is not precise enough to establish this information in an absolute sense. Furthermore, the seasonal distribution of snow fall is not known. See also the responses to reviewer 3 and the new Tab. S1 that entails the detailed sampling list (including the best estimate for the year and the sample averaging time).

- *Figure 2: I would like to see more than one background measurement.*

  We added more background freezing spectra to Fig. 2 (as well as the least active ice core freezing spectrum, as suggested by reviewer 1).

Literature

Beall, C. M., Lucero, D., Hill, T. C., DeMott, P. J., Stokes, M. D., and Prather, K. A.: Best practices for precipitation sample storage for offline studies of ice nucleation, Atmos. Meas. Tech. Discuss., https://doi.org/10.5194/amt-2020-183, in review, 2020.

Groot Zwaaftink, C. D., Grythe, H., Skov, H., and Stohl, A.: Substantial contribution of northern high-latitude sources to mineral dust in the Arctic, J. Geophys. Res. Atmos., 121, 13678–13697, https://doi.org/10.1002/2016JD025482, 2016.

Hartmann, M., Blunier, T., Brügger, S. O., Schmale, J., Schwikowski, M., Vogel, A.,Wex, H., Stratmann, F.: Variation of ice nucleating particles in the European Arctic over the last centuries, Geophysical Research Letters, 46, 4007–4016, https://doi.org/10.1029/2019GL082311, 2019.

Neff, J. C., Ballantyne, A. P., Farmer, G. L., Mahowald, N. M., Conroy, J. L., Landry, C. C., Overpeck, J. T., Painter, T. H., Lawrence C. R., and Reynolds, R. L.: Increasing eolian dust deposition in the western United States linked to human activity, Nature Geosci, 1, 189–195, https://doi.org/10.1038/ngeo133, 2008.

O'Sullivan, D., Murray, B., Ross, J., Whale, T. F., Price, H. C., Atkinson, J. D., Umo, N. S., and Webb, M. E.: The relevance of nanoscale biological fragments for ice nucleation in clouds, Scientific Reports, 5, 8082, https://doi.org/10.1038/srep08082, 2015.

Pongratz, J., Reick, C., Raddatz, T., and Claussen, M.: A reconstruction of global agricultural areas and land cover for the last millennium, Global Biogeochem. Cycles, 22, GB3018, https://doi.org/10.1029/2007GB003153, 2008.

---

## Author Comment (AC3) · 9 Sep 2020

*Review of Ice nucleating particle concentrations of the past: Insights from a 600 year old Greenland ice core*

*In this study Schrod et al, present the ice nucleating particle (INP) concentrations from a Greenland ice core spanning the past 600 years. The collected data set shows that the concentration of INPs has been rather consistent over the past 600 years. However, since 1960, the concentration and variability in INPs has increased. This has led the authors to suggest that human activities may be influencing INP concentrations, which could have significant impacts on future cloud radiative forcing. I appreciate that the authors are very careful in not over interpreting their results and are very thorough in addressing potential issues with conversions and contamination. I support the publication of this manuscript and provide some minor technical revisions. Additionally, I think it would be very interesting to extend the analysis to investigate the role of changing atmospheric circulation and rising arctic temperatures may have on the observed changes in INP concentration in this ice core sample.*

*General comments:*

*Although all layers of the ice core were treated the same and likely experienced similar temperature variabilities while accumulating on the ice sheet, it would be worthwhile to mention the recently found impacts of the storage on INPs relative to freshly collected samples. For example see Beall et al., (2020) and Stopelli et al., (2014). As the long term storage of the INPs in the ice may contribute to the observed difference between the ice core samples and precipitation samples shown in (Petters and Wright, 2015).*

> We thank the referee for directing our attention to potential storage effects. In fact, all reviewers agree that sample storage may have an important impact to the INP activity of the ice core samples. We now address this effect on several instances throughout the manuscript:

Page 5, line 11: "Hence, it was ensured that samples remained frozen at all times in our laboratory. However, sample vials may have been subject to temperatures between 0 °C and room temperature for up to some tens of hours in total (during repeated cycles of melting, storage and refreezing, non-INP measurements, and transport, etc.)."

Page 9, line 30: "However, recent studies indicate that sample storage (i.e. storage temperature) significantly affects the ice nucleation activity of fresh precipitation samples in the range of -7 °C to -19 °C (Beall et al., 2020). For example, samples stored at room temperature lost on average 72% of their INPs compared to the freshly analyzed samples. An average INP loss of 25% was still observed, even when samples were stored at -20 °C. Storage time did only weakly affect the INP concentrations. Therefore, based on this study a loss of INP activity on the order of a factor of 2 − 5 is possible, if not likely for the ice core measurements presented here. Furthermore, it is likely that the warmer end of INPs were disproportionally affected by these disturbances, while cold-temperature INPs were likely more robust. However, as all the samples experienced the same sample history, relative changes within the ice core can still be interpreted."

Page 10, line 26: "Hartmann et al. (2019) come to the same conclusion that INPs are well preserved in an ice core and a reconstruction of their concentration for past climates is possible. However, as previously stated, storage conditions may have affected the INP activation."

Page 16, line 5: "Considering that the total global agricultural land area is estimated to have increased by a factor of 10 from 1400 to 1992 (Pongratz et al., 2008) combined with the fact that wind erosion has immensely accelerated within the last two centuries (Neff et al., 2008), partly due to intensive grazing by the heavily increasing number of domesticated animals, one could even have expected larger differences between the two data groups. Especially in the temperature range around -15 °C, at which soil dust INPs from fertile agricultural regions are known to be active (O'Sullivan et al., 2015). We can only speculate why we generally did not observe many INPs in this temperature range, and why the significant differences between the two data groups were only observed for temperatures below -22 °C. First, it is possible that dust from anthropogenic practices was not transported to Central Greenland in a detectable amount. According to Groot Zwaaftink et al. (2016), most of the dust input contributing to the dust surface concentration of the Arctic is from Eurasia north of 60° N, North America north of 60° N and Asia south of 60° N. In contrast, North America and Europe south of 60° N, where land-use change and the agricultural expansion are most prominent, contribute only little to the Arctic dust input (below 1%). Moreover, Asian agricultural dust sources may not exhibit the necessary high wind speeds to inject mineral dust into the upper troposphere as required for long-range transport to Greenland. In contrast, mineral dust from the Taklamakan desert is intrinsically linked to dust storms in this area.

Second, the more fragile (biological) INPs may have been deteriorated during sample storage (Beall et al., 2020). As a result, the warm-end of INPs might have been largely lost, leaving only a homogeneous fraction of very stable INPs behind. Figures 8 and 9 present some evidence for this hypothesis. As seen in Fig. 8, we find a much narrower range of frozen fractions for the 10 year samples, hinting at a rather homogenous population of INPs. On the other hand, the variability is much higher for the modernday samples, possibly because some of the more fragile INPs were still active. However, as both sample groups experienced the same sample history after coring, this hypothesis would only be reflected by deterioration effects related to the time elapsed since the particles were deposited in the ice. Furthermore, Fig. 9 depicts increasingly greater relative differences in the INP concentration from -30 °C to -24 °C until the warmer end of the data is reached, at which only few samples show ice nucleation activity. This observation could possibly be explained by assuming that the warmer INPs were largely deactivated due to storage effects."

*As each of the samples used to probe the concentration of INPs every 10 years only covers a period of 6 months, is the 6 month period roughly the same for each of the 10 yr samples? Based on Fig. 6, the variability over a year (monthly sample from 1463-64) looks to be about an order of magnitude. Therefore, if the 6 months covered by a 10 yr sample differs, some of the variability between the 10 yr samples, albeit a small amount, could be explained.*

Theoretically, we selected the sample vials for the 10 year time series, whose midpoints were closest to the same season of a year (e.g. 1950.0). However, the uncertainty in the dating of the ice core effectively does not allow to assume that each sample corresponds to the same season (also the "sample resolution" varies to some degree, see Fig. 1b). Therefore, we agree to the referee that some part of the overall variability can be explained by possibly comparing different times of a year. We added a table to the supplement entailing the sample list (Tab. S1; sample number, depth, estimated year, representative time average, data subset). Also see the following answer. We added a sentence to page 6, line 9:

"Whenever possible, we selected samples that theoretically represented the same season(s). However, due to the uncertainty in the ice core dating some of the variability in the INP concentration may be attributed to seasonal differences."

*The same question is also relevant for the modern day samples (Fig. 7) where there are some years with higher activity than others. It would be important to know if the yearly samples (actually only 6 months) cover the same 6 month period for each year.*

The same procedure was applied to the modern day samples. However, due to the higher sampling frequency of the modern day samples the number of samples to choose from was lower. Accordingly, there is somewhat more diversity to be expected for this group with regards of the averaged seasons. We added a few words to page 15, line 10:

"Furthermore, the results could potentially be intrinsically influenced to some degree by differences in sampling frequency and time coverage, as well as samples representing different seasons of the year."

On a related matter we now added an alternative figure to the Supplement (Fig. S2), similar to Fig. 7, but with the original (non-interpolated) time coverage:

[Figure]

*Here it is shown that the Anthropocene samples are significantly different from the preindustrial samples. This is a very interesting finding and something that the authors suggest may be due to a change in the dust due to desertification, and other anthropogenic related aerosols that reach the Greenland ice sheet. Although these seem like possibilities, it would be interesting to discuss the potential influence from changes in atmospheric circulation patterns such as the NAO (Pinto and Raible, 2012).*

We regard changes in the emission strength much more likely than changes to atmospheric circulation patterns. To our knowledge, there is no clear evidence from aerosol tracers or models suggesting that there were strong anthropogenic circulation changes in Greenland yet. However, some changes in transport patterns cannot be entirely disregarded. In fact, we address this on page 10, line 4 – 8. Related effects on dry and wet deposition, which may be caused by a change in atmospheric transport patterns, and the relevance to the interpretation of the INP results are described on page 11, line 20 – 24. Also see following answer.

*Additionally, it has been shown that precipitation effectively removes precipitation (Stopelli et al., 2015) and as the ice core site is at a high altitude arctic site, it may be extremely sensitive to the temperature and amount of precipitation that falls (removal of INPs) upstream of the site. The fact that an overall increase in IN activity has been observed in more recent, warmer years may be consistent with warmer air masses precipitating over the ice sheet where fewer INPs have been removed upstream compared to previous (colder) years. Therefore, it may be worthwhile to compare the INP concentrations with the reconstructed temperature record over the same period from the ice core.*

We assume the first sentence was meant to read: "it has been shown that precipitation effectively removes INPs" (?). We recognize that temperature and precipitation are two dominant factors determining the deposition efficiency and therefore the amount of INPs in the accumulated snow (see sections 2.9 and 2.10). The suggested idea to compare the warming climate via a reconstructed temperature record to the INP concentration is definitely an interesting thought. Yet, we feel it is beyond the scope of this manuscript and could be explored further in a follow-up study, which might include a modelling aspect.

We added a paragraph to the discussion in section 3.1 including the last two points of the referee:

"Now the question arises, what factors may have caused these significant differences in INP concentrations. Several hypothetical explanations come to mind. First, the changing climate may have influenced both the deposition pathways and their efficiencies (cf. sections 2.9 and 2.10). But, at least locally, the accumulation rate at B17 does not show a change between modern and pre-industrial times. Further, changes to relevant large-scale atmospheric circulation patterns are essentially unknown for the investigated time period. [...]"

*Minor comments:*

- *Page 3, line 3: Consider adding the following references: Grawe et al., (2016, 2018); Kanji et al., (2020); Ullrich et al., (2016)*

    Okay.

- *Page 3 line 14: Consider adding the following references: Hill et al., (2016); Steinke et al., (2016)*

    Okay.

- *Page 3 line 20: It is highlighted here that the dominant dust sources in Greenland ice cores come from Chinese deserts and the Taklamakan. Therefore, it would be worthwhile to discuss the observed ability of these mineral dusts to act as INPs. Do they match in terms of INA with the observed INPs found in the ice cores (it seems like they do)? Consider mentioning previous studies on INPs from this region such as Boose et al., (2016); Field et al., (2006); Paramonov et al., (2018); Ullrich et al., (2016).*

    We agree that a short discussion about the ice nucleation activity of the mineral dust from the relevant desert regions is an interesting addition. We thank the reviewer for the suggested literature. The manuscript now reads:

    "Mineral dust from China, and the Taklamakan desert in particular, has been characterized in several laboratory ice nucleation studies (Field et al., 2006; Niemand et al., 2012; Boose et al., 2016; Ullrich et al., 2017; Paramonov et al., 2018), which revealed a relatively high ice active site density in the temperature range below -25 °C, comparable to other natural deserts such as the Sahara (e.g. Niemand et al., 2012; Boose et al., 2016)."

    It is, however, difficult to assess how well these laboratory measurements with pure dust from the Chinese deserts match to the observed INP spectra in the ice core. Most times, laboratory studies provide the active site density $n_s$ as their metric for ice nucleation activity. We don't have a good enough characterization of aerosol particles within the ice core samples to estimate a reliable $n_s$ value, which makes the comparison difficult. Assuming that all particles counted by the CFA measurements have a spherical diameter of 1.2 µm we can estimate the active site density to be on average $1.5 \pm 6.1 \times 10^{10}$ $m^{-2}$ (range: $1.4 \times 10^8 - 6.3 \times 10^{11}$ $m^{-2}$) at -25 °C and $8.4 \pm 13.3 \times 10^{10}$ $m^{-2}$ (range: $4.4 \times 10^9 - 1.0 \times 10^{12}$ $m^{-2}$) at -30 °C. Obviously, this assumption is flawed, as there will be smaller particles that could not be counted by the CFA measurements, which would add to the total aerosol surface area, as well as

particles that were larger than 1.2 µm, which were here assumed to have this lower size. However, the estimated $n_s$ values are indeed in the range of those presented in the literature for the Taklamakan desert and other Chinese deserts. We added a paragraph to section 3.4:

"Moreover, evidence presented in section 3.3 and Tab. 1 indicated that the long-range transported dust from East Asian deserts influenced the freezing characteristics of the ice core samples. Laboratory studies characterizing the ice nucleation activity of mineral dust from the Taklamakan desert and other Chinese deserts report active site densities ns at -25 °C of approximately $1 \times 10^{10}$ m$^{-2}$ (Niemand et al., 2012; Ullrich et al., 2017) and between $1 \times 10^{10}$ to $1 \times 10^{11}$ m$^{-2}$ at -30 °C (Niemand et al., 2012; Boose et al., 2016; Ullrich et al., 2017; Paramonov et al., 2018). We can only roughly calculate $n_s$ from the CFA particle measurements. Lacking a solid particle size distribution measurement, we assumed all counted particles to have a spherical diameter of 1.2 µm. This assumption is obviously flawed, as particles smaller than 1.2 µm were not counted by the CFA measurements, and larger particles were assumed to have this lower size. With this rough assumption, we find an average $n_s$ of $2 \pm 6 \times 10^{10}$ m$^{-2}$ at -25 °C and $8 \pm 13 \times 10^{10}$ m$^{-2}$ at -30 °C, which is in surprisingly good agreement with the literature."

- *Page 6 line 14: change "must" to "does"*
  Okay.

- *Page 6 line 20-21: why was the seasonal variability explored in the 1463? Is there a reason for choosing this period? Wouldn't a more recent year make it easier to identify the months of the year as the ice is less compact?*
  There was no particular reason why the year 1463 was explored specifically other than opportunity. The INP analysis was done after CFA and IC measurements were already performed. Particularly, the CFA decontamination step determined the resolution of the samples. There were only two periods with samples of near monthly resolution (average: approx. 10 samples per year) from which seasonal cycles could be established: 1744 – 1763 and 1454 – 1468). The latter period was chosen for being unaffected by the industrial revolution. The exact samples were chosen more or less at random. (The chosen samples were in the center of the period and preliminary CFA data showed a clear annual signal.) But we agree with the reviewer that it would be very interesting to investigate the seasonal resolution of more recent years in future studies (also see page 19, lines 2 – 4). We added a sentence to page 16, line 3:

  "There was no particular reason why this year was explored specifically. As the CFA decontamination step determined the resolution of samples, there were two periods with samples of near monthly resolution from which seasonal cycles could be established (1744 – 1763 and 1454 – 1468). The latter period was chosen for being unaffected by the industrial revolution."

- *Page 6 line 28: "hast" should be "has"*
  Okay.

- *Page 7 line 5: consider rephrasing "picked up" to "pipetted"*
  Okay.

- *Page 7 line 7-8: Why is FRIDGE kept at 14 C initially? Based on what was stated earlier, the samples were defrosted at 6 C, so why wasn't FRDIGE set to 6 C to minimize the temperature range a sample was exposed to. Granted, all of the samples experienced the same treatment so this likely has no impact on the overall comparison between samples.*

  The initial temperature was set to 14 °C for practical reasons and is based on the experience of the operators. While pipetting the droplets, the chamber needs to be partially opened. When the chamber is open the flow of dry synthetic air did sometimes not suffice to prevent condensation on the surface of the wafer substrate if the temperature was set to a cooler temperature (depending on ambient conditions). Overall, we don't think that the short amount of time during pipetting altered the freezing spectra substantially. (However, other storage effects may be more relevant, see other responses and specifically the comments of reviewer 2.)

- *Page 7 line 8-10: Do you mean that the Lauda cryostat was used to dissipate heat from the Peltier element. Please rephrase this sentence to make that clearer.*

  Yes. The manuscript now reads:
  "The temperature ramp was implemented by a PID-controlled Peltier element. A cryostat (Lauda, Ecoline Staredition RE110; ethanol coolant) was used to dissipate the heat from the Peltier element."

- *Page 7 line 11: Does the synthetic air flush change the size of the droplets during the experiment via evaporation? If yes, would this be significant enough to increase the concentration of solutes in a droplet such that it may lead to a freezing point depression in the samples? In theory, the colder the cell gets (the longer the experiment lasts) the more concentrated these solutes would become.*

  Judging from measurement images we can't say for sure if or how much of the droplet size is shrinking due to evaporation. As for freezing point depression, CFA and IC measurements indicate that solutes such as $Na^+$ and $Ca^{2+}$ are on the order of 10 to 100 ng/mL, which is very low. We agree to the referee that the possible effect of a freezing point depression will increase with shrinking droplet size, however we don't think this will have a significant effect on the overall results.

- *Page 7 line 18: Here you mention mL of meltwater but then use mLice when reporting INP concentrations. Consider making the terminology consistent.*

  Okay.

- *Page 8 line 6: Why was the SEM analysis conducted on the samples after being filtered (400 nm pore size) when the highest correlation between INP concentration and particles concentrations was for particles larger than 1.2 microns? Do these large particles make it through the filter?*

  We believe, the reviewer misunderstood the SEM preparation procedures: The sample water was pumped through a 400 nm pore size filter using a water jet vacuum. Then these filters were then analyzed with SEM. Large particles will therefore be present during the SEM analysis, whereas smaller particles will be lost.

- *Page 8 line 27: Check if "microscopical" should be "microscopic" in this case.*

  Okay.

- *Page 9 line 29: Here it is mentioned that the freezing and melting of the same droplets does not influence the ice nucleating ability of the samples. As previously mentioned in the general comments, it might be worthwhile to mention other studies where it was shown that over longer periods, the storage and repeated melting and freezing of samples influenced the ice nucleating ability of samples.*

  We thank the reviewer for pointing out the possible deactivation of INPs by storage, etc. We will add to the manuscript at the suggested lines (and elsewhere):

  "However, recent studies indicate that sample storage (i.e. storage temperature) significantly affects the ice nucleation activity of fresh precipitation samples in the range of -7 °C to -19 °C (Beall et al., 2020). For example, samples stored at room temperature lost on average 72% of their INPs compared to the freshly analyzed samples. An average INP loss of 25% was still observed, even when samples were stored at -20 °C. Storage time did only weakly affect the INP concentrations. Therefore, based on this study a loss of INP activity on the order of a factor of 2 – 5 is possible, if not likely for the ice core measurements presented here. Furthermore, it is likely that the warmer end of INPs were disproportionally affected by these disturbances, while cold-temperature INPs were likely more robust. However, as all the samples experienced the same sample history, relative changes within the ice core can still be interpreted."

- *Page 12 line 22: Remove extra "/" after gprecip in first term of equation*

  Okay.

- *Page 14 line 8: please specify that this is the concentration at -20 C as mention of -20C comes two sentences earlier.*

  Actually, page 14, line 8 gives the INP concentration at -25°C, which is mentioned two sentences earlier. We believe the structure of the paragraph is clear without repeating the temperature in every sentence.

- *Page 16 line 10-11: How do these large particles make it through the 400 nm pore sized filters described in the methods?*

  See comments above.

- *Page 16 line 27: Here it is mentioned that there is a seasonal cycle in INP and although the variability is significantly less than the over the entire period of the study, it may be worth mentioning if the 6 month samples are taken to over the same 6 months in every time point (as said in the general comments).*

  See comments above.

- *Page 17 line 15-25: Could some of the differences in the INP concentrations be due to the droplet size used in the studies? Perhaps the small droplet volume in this study makes the measurement of rarer INPs less quantifiable. Additionally, could location differences between sampling sites, lead to differences in the number and efficiency of INPs removed upstream of the sites (Stopelli et al., 2015). For example, Svalbard often experiences periods of relatively warm air masses laden with INPs that would*

*precipitate out before reaching the high altitude location of this core. These points,*
*although briefly mentioned, could be expanded on.*

Drop size used in Hartmann et al. (2019) was 1 µL (LINA) and 50 µL (INDA). Our droplet size of 2.5 µL was therefore somewhere in the middle of those. We therefore think that the droplet size should not have biased the comparison greatly. Of course, the droplet size determines the effectively observed freezing range, but in the Vali (1971) equation of the cumulative concentration it is accounted for. However, generally speaking we agree to the reviewer that a small droplet volume might make it less likely for rare INPs to be quantifiable.

Geographical differences on the INP number and deposition efficiency between the core sites are well within the realm of possibility. Additionally, we now add possible storage effects as a further possible reason for the different concentration range observed. We expanded upon the lines, which now read:

"This disparity may arise from experimental (droplet volume, etc.), methodological (e.g. sample storage conditions) and or geographical differences, which may affect the deposition mechanisms and efficiency."

Literature

Beall, C. M., Lucero, D., Hill, T. C., DeMott, P. J., Stokes, M. D., and Prather, K. A.: Best practices for precipitation sample storage for offline studies of ice nucleation, Atmos. Meas. Tech. Discuss., https://doi.org/10.5194/amt-2020-183, in review, 2020.

Boose, Y., Welti, A., Atkinson, J., Ramelli, F., Danielczok, A., Bingemer, H. G., Plötze, M., Sierau, B., Kanji, Z. A., and Lohmann, U.: Heterogeneous ice nucleation on dust particles sourced from nine deserts worldwide – Part 1: Immersion freezing, Atmos. Chem. Phys., 16, 15075–15095, https://doi.org/10.5194/acp-16-15075-2016, 2016.

Groot Zwaaftink, C. D., Grythe, H., Skov, H., and Stohl, A.: Substantial contribution of northern high-latitude sources to mineral dust in the Arctic, J. Geophys. Res. Atmos., 121, 13678–13697, https://doi.org/10.1002/2016JD025482, 2016.

Field, P. R., Möhler, O., Connolly, P., Krämer, M., Cotton, R., Heymsfield, A. J., Saathoff, H., and Schnaiter, M.: Some ice nucleation characteristics of Asian and Saharan desert dust, Atmos. Chem. Phys., 6, 2991–3006, https://doi.org/10.5194/acp-6-2991-2006, 2006.

Hartmann, M., Blunier, T., Brügger, S. O., Schmale, J., Schwikowski, M., Vogel, A.,Wex, H., Stratmann, F.: Variation of ice nucleating particles in the European Arctic over the last centuries, Geophysical Research Letters, 46, 4007–4016, https://doi.org/10.1029/2019GL082311, 2019.

Neff, J. C., Ballantyne, A. P., Farmer, G. L., Mahowald, N. M., Conroy, J. L., Landry, C. C., Overpeck, J. T., Painter, T. H., Lawrence C. R., and Reynolds, R. L.: Increasing eolian dust deposition in the western United States linked to human activity, Nature Geosci, 1, 189–195, https://doi.org/10.1038/ngeo133, 2008.

Niemand, M, Möhler, O., Vogel, B., Hoose, C., Conolly, P., Klein, H., Bingemer, H., DeMott, P., Skrotzki, J., and Leisner, T.: A Particle-Surface-Area-Based Parameterization of Immersion Freezing on Desert Dust Particles, J. Atmos. Sci., 69, 3077–3092, https://doi.org/10.1175/JAS-D-11-0249.1, 2012.

O'Sullivan, D., Murray, B., Ross, J., Whale, T. F., Price, H. C., Atkinson, J. D., Umo, N. S., and Webb, M. E.: The relevance of nanoscale biological fragments for ice nucleation in clouds, Scientific Reports, 5, 8082, https://doi.org/10.1038/srep08082, 2015.

Paramonov, M., David, R. O., Kretzschmar, R., and Kanji, Z. A.: A laboratory investigation of the ice nucleation efficiency of three types of mineral and soil dust, Atmos. Chem. Phys., 18, 16515–16536, https://doi.org/10.5194/acp-18-16515-2018, 2018.

Pongratz, J., Reick, C., Raddatz, T., and Claussen, M.: A reconstruction of global agricultural areas and land cover for the last millennium, Global Biogeochem. Cycles, 22, GB3018, https://doi.org/10.1029/2007GB003153, 2008.

Ullrich, R., Hoose, C., Möhler, O., Niemand, M., Wagner, R., Höhler, K., Hiranuma, N., Saathoff, H. and Leisner, T.: A New Ice Nucleation Active Site Parameterization for Desert Dust and Soot, J. Atmospheric Sci., 74, 699–717, https://doi.org/10.1175/JAS-D-16-0074.1, 2017.

Vali, G.: Quantitative Evaluation of Experimental Results an the Heterogeneous Freezing Nucleation of Supercooled Liquids, J. Atmos. Sci., 28, 402–409, https://doi.org/10.1175/1520-0469(1971)028<0402:QEOERA>2.0.CO;2, 1971.